# Compound Flood Events: Analysing the joint occurrence of extreme river discharge events and storm surges in Northern and Central Europe

Philipp Heinrich[1], Stefan Hagemann[1], Ralf Weisse[1], Corinna Schrum[1,2], Ute Daewel[1], and Lidia Gaslikova[1]

[1]Helmholtz-Zentrum Hereon, Max-Planck-Straße 1, 21502 Geesthacht, Germany
[2]Institute of Oceanography, University of Hamburg, Bundesstraße 53, 20146 Hamburg, Germany

**Correspondence:** Philipp Heinrich (philipp.heinrich@hereon.de)

**Abstract.** The simultaneous occurrence of extreme events gained more and more attention from scientific research in the last couple of years. These compound extreme events have an increased damage potential, compared to the separate occurrence of extreme events, and became a core topic in risk assessment. It is important to improve our understanding of the underlying mechanisms that can cause compound flood events. Our study focuses on the analysis of compound flood events with the following contributions. First, we introduce a Monte–Carlo approach to analyse flood event probabilities in northern and central Europe without the use of copulas or ensembles. Second, we investigate if the number of observed compound extreme events is within the expected range of two standard deviations of randomly occurring compound events. This includes variations of several parameters to test the stability of the identified patterns. Finally, we analyse if the observed compound extreme events had a common large-scale meteorological driver. The results of our investigation show that rivers along the western facing coasts of Europe experienced a higher amount of compound flood events than expected by pure chance. In these regions, the vast majority of the observed compound flood events seem to be related to the Großwetterlage Cyclonic Westerly.

## 1 Introduction

Coastal flooding is one of the most frequent, expensive, and fatal natural disasters. In the U.S. alone, it dealt $199 billion in flood damages from 1988 to 2017 according to Davenport et al. (2021). For Europe, Vousdoukas et al. (2018) projected an increase of annual costs caused by coastal floods of up to $1 trillion in 2100 for representative concentration pathway (RCP) RCP8.5. Furthermore, more than 600 million people live in coastal areas that are less than 10 meters above sea level and less than 100 kilometres from the shore (United Nations, 2017; McGranahan et al., 2007). Drivers for floods are storm surges, waves, tides, precipitation, and high river discharge (Paprotny et al., 2020). Additionally, floods can also be the result of failures of critical infrastructure like hydropower dams or flood defences (ECHO, 2021).

The IPCC Special Report on Managing the Risks of Extreme Events and Disasters to Advance Climate Change Adaptation (SREX) defined compound events as "(1) two or more extreme events occurring simultaneously or successively, (2) combinations of extreme events with underlying conditions that amplify the impact of the events, or (3) combinations of events that

are not themselves extremes but lead to an extreme event or impact when combined. The contributing events can be of similar (clustered multiple events) or different type(s)" (Seneviratne et al., 2012). A more general definition was proposed by Leonard et al. (2014), who defined it as "an extreme impact that depends on multiple statistically dependent variables or events." Compound flood events occur when large runoff from, e.g., heavy precipitation, leading to extreme river discharge, is combined with high sea level (storm surge).

The occurrence of extreme flood events either simultaneously or in close succession can lead to severe damage, which greatly exceeds the damage those events would cause separately (de Ruiter et al., 2020; Xu et al., 2022). Several studies conducted over the last years have shown the importance and catastrophic nature of compound flood events for several locations. Examples for these are compound inland floods of 2013 and 2016 in Germany (Thieken et al., 2022), the Siret Basin flood in northeastern Romania in 2010 (Romanescu et al., 2018), and the study on compound flooding in Houston-Galveston Bay during Hurricane Harvey (Valle-Levinson et al., 2020). Moreover, several studies have been conducted on a larger spatial scale in Europe, which cover North-western Europe (Ganguli and Merz, 2019; Ganguli et al., 2020), Norway (Poschlod et al., 2020), eastern Britain (Svensson and Jones, 2002), and entire Europe (Bevacqua et al., 2019; Paprotny et al., 2018b, 2020). Those studies investigated possible dependencies, mostly between surge and discharge. All of them found that the assumption of independence between drivers leads to a huge underestimation of the occurrence rate of compound events.

In addition to the large-scale studies mentioned above, a large number of studies exist that focus on smaller regions. Examples are the studies of van den Hurk et al. (2015) and Santos et al. (2021a), which both analysed a near flood event in the Netherlands in January 2012, which was caused by a combination of extreme weather conditions. Studies have been conducted worldwide, with examples being the Zengwen River basin in Taiwan by Chen and Liu (2014), Shoalhaven River in Australia by Kumbier et al. (2018), Fuzhou in China by Lian et al. (2013), Dickinson Bayou watershed in Texas by Kim et al. (2022), and other locations in various countries.

It is impossible to compare the results of different publications directly, since every study uses different approaches and measurements. There are currently no established standards for detecting extreme events. For example, the thresholds for extreme events were calculated by utilising the return period (Bevacqua et al., 2019), a certain number of events per year (Hendry et al., 2019; Ganguli et al., 2020), or utilising a percentile approach (Paprotny et al., 2018a). Other studies chose block maxima to detect extreme events (Engeland et al., 2004). The exact parameters are chosen nearly arbitrarily by the authors, with the only common goal being a low number of events so that they can be declared as 'extreme'. Nonetheless, there have been some studies that investigated the sensitivity of their results regarding different methods of identifying extreme events (Zheng et al., 2014) and changes to the model set-up (Jane et al., 2022). Additionally, there are studies like Ghanbari et al. (2021) that compare their results to those of other studies, e.g., to Ward et al. (2018). Basically, all studies found a correlation between drivers to a certain extent.

The influence of climate change on the frequency of compound flood events in Europe has been investigated by different studies. The increasing sea level due to climate change and higher occurrence of strong precipitation pose an increasing threat to important economic centres around the world and the people living there (Müller and Sacco, 2021). Feyen et al. (2020) projected that in case of a high emissions scenario, the damages caused by floods will represent a considerable proportion of

some country's national gross domestic product (GDP) at the end of the century. Studies that investigated the effect of climate change on compound flood events focused on various regions of interest, for example, Bevacqua et al. (2019) on entire Europe, Poschlod et al. (2020) on Norway, Bermúdez et al. (2021) on the rivers Mandeo and Mendo in Spain, and Ganguli et al. (2020) on northwestern Europe. Bevacqua et al. (2019, 2020) reported a strong increase in the occurrence rate of compound flooding events for the future, especially for northern Europe, mainly due to the stronger precipitation as the result of a warmer atmosphere carrying more moisture. Contrarily, Ganguli et al. (2020) reported a lower risk of compound flooding due to a lower dependence between surges and river discharge peaks.

Many studies utilised copulas to describe the data distribution of two time series and investigate the dependence between extreme events. They were introduced in Sklar (1959). The Upper Tail Dependence Coefficient (UTDC) is used to estimate the probability of two extreme events occurring at the same time and to derive their level of dependence. It can be calculated, for example, with the Capéraá–Fougéres–Genest estimator (Capéraà et al., 2000). However, copulas introduce an additional amount of uncertainty (Heffernan and Tawn, 2004) and may require a large number of data points for robust tail dependence calculation (Moran, 1957). In climate research, the amount of available data points is often too small for this kind of analysis with many studies operating at merely 30 extreme events. Serinaldi et al. (2015) therefore concluded that those results are "highly questionable and should be carefully reconsidered".

Using a thorough mathematical analysis, Frahm et al. (2005) showed that the calculation of the Upper Tail Dependence Coefficient is based on the assumption that an actual correlation between the datasets exists and therefore will always have a strong bias. This goes to the point that Joe (2014) stated that "the empirical measure of tail dependence for data doesn't really exist". This raises the question of how to quantify the number of compound flood events and the resulting dependence between their drivers. Consequently, we chose to study compound flood events by using a methodology that does not utilise copulas.

In the present study, we analyse compound flood events by focusing on the question whether they occur more often than by pure coincidence. Utilising several available large-scale data sets allowed us to conduct this analysis for Northern and Central Europe, instead of focusing on a single river. Furthermore, we wanted to investigate if spatial patterns occur and if they are caused by one common meteorological driver. To achieve this, we implemented a simple statistical method that avoids the application of copulas. For this, we randomised our datasets and investigated if the resulting random distributions show comparable amounts of compound extreme events. Similar studies were so far only carried out by van den Hurk et al. (2015) for the Lauwersmeer in the Netherlands and by Poschlod et al. (2020) for Norway (in this case covering rain on snow events). To our knowledge, this will be the first recent publication investigating compound flooding in northern Europe without the use of copulas. For this, we utilised discharge and sea level datasets that were simulated based on reanalysis and hindcast data. Moreover, we investigated the robustness of the spatial patterns in our results by modifying various parameters of our method, like the thresholds for determining extreme events. Additionally, we investigated potential correlations between a river's catchment size and the number of compound flood events that occur. Finally, we examined possible drivers that could cause the occurrence of compound flood events.

## 2 Methods

The first step in determining extreme events is to define which events are considered to be 'extreme'. There are ways to use automatic threshold approaches for detecting extreme events, like goodness of fit p-value (Solari et al., 2017) or the characteristics of extrapolated significant wave heights (Liang et al., 2019), but they struggle due to the diverse characteristics in the time series of drivers that cause coastal floods (Camus et al., 2021). River-specific thresholds are only feasible for case studies that can take the local properties, like flood protection or elevation of the surrounding area, into account. Therefore a more general approach is needed that is applicable to all rivers. As described in Sect. 1, there is so far no standardised method that is generally used. Quite the contrary, every study uses its own modus operandi, each having individual reasoning for their choice.

One option is utilising block-maxima for extreme event detection (Gumbel, 1958), which provides a well-spaced distribution of extreme events, e.g. one event per year, meaning one annual maximum event. However, it can miss out on events with high values, in case several events happen in the same year (Santos et al., 2021b), while also labelling lower values as 'extreme' in years without any major events.

We, therefore, chose the peaks-over-threshold (Pickands III, 1975) method to select extreme events by using percentiles, like in the works of Rantanen et al. (2021), Fang et al. (2021), Lai et al. (2021), Ward et al. (2018), and Ridder et al. (2018). While using the peaks-over-threshold method, it is important to ensure the independence of the events. It has to be prevented that, for example, a single day that slightly drops under the thresholds, creates two separate events (Harley, 2017). For this reason, we chose a de-clustering time of three days. This means that two events are considered to be separate if the threshold is not exceeded for four consecutive days, like in Haigh et al. (2016).

Furthermore, the choice of our threshold needed to take the limited data availability into account. Hence, we were forced to choose our thresholds low enough to ensure that enough points were available for robust statistical analysis. The number of extreme discharge events can vary strongly depending on the river itself. Large rivers like the Elbe show the tendency of having very long extreme events that can last for several weeks, therefore resulting in a comparably lower number of extreme events. Smaller rivers, however, have usually rather short extreme events, consequently having most of the time higher amounts of them. While this specific approach might result in nominally different numbers of extreme events for each river, it ensures that for each river the same amount of data points exceed the threshold. Sea level also exhibits variations in event duration, albeit to a lesser extent. For the discharge of rivers we chose the $90^{\text{th}}$ percentile $D_{90}$ and for the sea level the $99^{\text{th}}$ percentile $S_{99}$.

In order to enable a good comparison between different rivers, the number of extreme discharge and sea level data points should be the same for all of them. Then again, extreme events should be rare by definition, regardless of the river size, therefore only occurring scarcely throughout the year. This especially prevents the accidental analysis of events that are normally not considered to as extreme.

To test the influence of the extreme event definition on possible patterns, we additionally implemented an automatic threshold tuning that modifies the percentiles and the subsequent thresholds in such a way that they result in an average of two extreme events per year. This was done to test in Sect. 4.2 if our results remain stable under much stricter definitions of extreme events.

Moreover, the threshold tuning results in an average return period of 0.5 years for extreme discharge and sea level events since the return period can be defined as

$$RP = \frac{L}{E} \tag{1}$$

where $RP$ denotes the return period, $L$ the duration of the data set in years, and $E$ the number of extreme events.

Another factor we had to take into account is the so-called *lag*, which characterises the temporal delay between variables
reacting to the same meteorological event. Such lag can occur for example, if a storm approaches a coast it generates increased sea level due to stronger winds, before travelling inland where it causes higher amounts of discharge due to precipitation. Most studies, e.g. Hendry et al. (2019), tested a variety of ranges like $\pm$ five days, while Ganguli et al. (2020) calculated the delay based on the catchment size of the river.

There is a valid argument made by Ward et al. (2018) that the delay can put high stress on the flood protection systems if
the initial flood water can't retreat fast enough before the discharge occurs. Due to the large area of our study, it is impossible to quantify the potential consequences of ongoing floods on the coastal protection system for each river. Hence, we decided to focus on joint occurrences of extreme events without any additional lag. Despite that, we tested our results in Sect. 4.2 for a lag of 3 days to investigate potential influences on our results. In our case, we used the lag as a temporal search radius around the discharge extreme event rather than a shift of the time series itself.

To identify rivers that show a higher number of compound flood events than expected by pure chance, we utilised a Monte–Carlo approach. Other studies in the past also utilised data permutation, see, for example, Svensson and Jones (2002), Zheng et al. (2013), and Nasr et al. (2021). Rivers with this behaviour might indicate a common large-scale driver that causes extreme discharge and sea level at the same time. For example, Hendry et al. (2019) found that the compound events on the west coast of Great Britain have a different meteorological background than those on the east coast. A randomisation method was used
to disrupt possible correlations between the data sets and see how the number of compound flood events changes in the case of independent data. First, we limited the time frame of the data sets to the late fall and entire winter season, as storm surges mostly occur in the winter season in northern Europe, see for example (Liu et al., 2022). For the winter season, we used a time frame from December to February, such as also done by Robins et al. (2021). At the same time, most discharge events are also limited to the winter and early spring seasons. Neglecting this seasonality would naturally lead to false-positive dependencies
since seasonal events would be spread throughout the entire year instead of being mostly limited to their own season. As a result, we would see a much lower number of compound flood events in the non-randomised data; therefore suggesting a false dependence. This was similarly stated by Couasnon et al. (2020).

Afterwards, we determined the number of compound flood events by counting the joint occurrence of extreme events in the discharge and sea level data. To deal with the differing duration of discharge events, we counted the occurrence of multiple
separate sea level extreme events during the same discharge event as separate compound flood events.

After determining the number of compound flood events in the original data sets, we prepared the randomisation of the sea level data. For this, we made sure that events were not split up by grouping data points of the same event together before the shuffling process. This was done to not artificially increase the number of extreme events by separating events that consist of

more than a single data point. Every data point that was not an extreme event was put into its own group as the only member. The shuffling process of the groups with numpy (Harris et al., 2020) assigned every group a weight based on the number of data points inside each group. After the shuffling process, the groups were disbanded and formed a randomised data set based on the new order. Afterwards, we performed the de-clustering process again to ensure that extreme event data points in close proximity were counted as a single event. Then we calculated the number of compound flood events for the combination of discharge data and randomised sea level data. This bootstrap process was repeated 10,000 times for each river, giving a probability distribution for each of them. The resulting probability distribution was used to determine if the initially observed number of compound flood events is within the 95% confidence interval of two standard deviations ($2\sigma$).

To test the robustness of our results in Sect. 4.2, we also created an additional randomisation approach by randomly shuffling the order of the winter months throughout the sea level data. This method was easier to implement than the one used for the main analysis. For further testing, we utilised different combinations of datasets to investigate their influence on our results. Finally, we used two different time frames to see if climate change or the choice of time period have an influence on possible spatial pattern.

The domains of all catchments, regions, and seas that we mention by name for various reasons in this study can be seen in Fig. 1.

## 3 Data

In order to study compound flood events, spatial and temporal consistent long time series of daily river runoff (discharge) and sea level near the coast are required. On the one hand, observed discharges are usually not available at the respective river mouths, but they are often measured at stations further inland. In addition, periods of available daily data vary considerably between the rivers, even over the considered region that has a rather good data coverage.

Consequently, we chose several model-generated datasets that provide daily data also for sea level over a time period of at least 20 years and cover northern Europe. For our analysis, we utilised several data sets which varied in forcing, regions and time frames. This was done to enable robustness tests of our analysis under a diverse set of conditions. These datasets were generated by using observations and reanalysis data as forcing and they are described below. A short overview of their usage in this paper is given in Table 1.

### 3.1 River Runoff

We utilised two daily river runoff datasets that are based on consistent long-term reconstructions by the global hydrology model HydroPy (Stacke and Hagemann, 2021) and the hydrological discharge (HD) model (Hagemann et al., 2020). The river runoff was simulated at 5 Min. spatial resolution covering the entire European catchment region. The HD model Vs. 5.0 (Hagemann and Ho-Hagemann, 2021) was set up over the European domain covering the land areas between -11° W to 69° E and 27° N to 72° N at a spatial resolution of 5 min (ca. 8–9 km). Both datasets were published as Hagemann and Stacke (2021) and utilised in Hagemann and Stacke (2022).

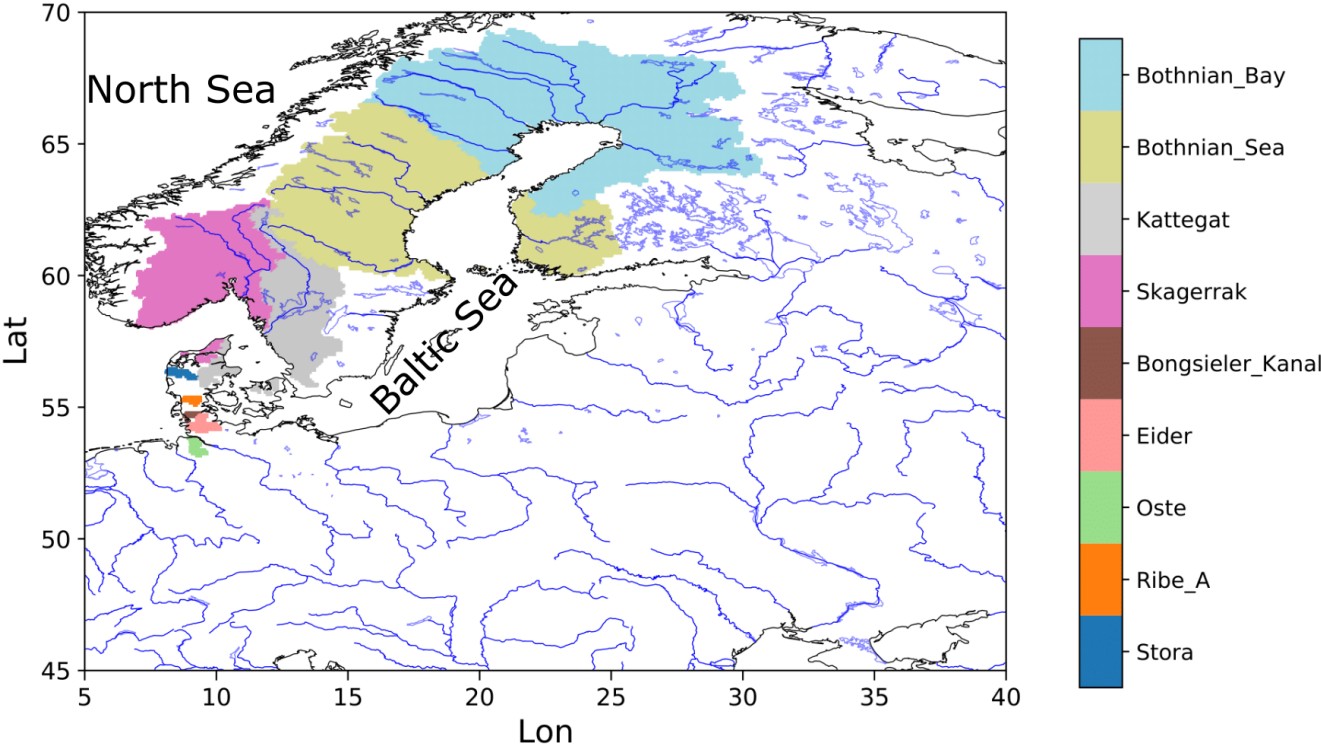

**Figure 1.** This figure contains the catchments, regions, and seas that are mentioned by name throughout the study. The first five entries in the colorbar contain maritime zones with highlighted catchment areas of rivers that discharge into them. The last five entries show the catchment area of five rivers on the German-Danish western coast.

### 3.1.1 HD5–ERA5

ERA5 is the fifth generation of atmospheric reanalysis (Hersbach et al., 2020) produced by the European Centre for Medium-Range Weather Forecasts (ECMWF). It provides hourly data on many atmospheric, land-surface, and sea-state parameters at about 31 km resolution. HydroPy was driven by daily ERA5 forcing data from 1979–2018 to generate daily fields of surface and subsurface runoff at the ERA5 resolution. Here, the Penman–Monteith equation was applied to calculate a reference evapotranspiration following Allen et al. (1998). Then, surface and sub-surface runoff were interpolated to the HD model grid and used by the HD model to simulate daily discharges.

### 3.1.2 HD5–EOBS

The E-OBS data set (Cornes et al., 2018) comprises several daily gridded surface variables at 0.1° and 0.25° resolution over Europe covering the area 25° N–71.5° N × 25° W–45° E. The data set has been derived from station data collated by the ECA&D (European Climate Assessment & Dataset) initiative (Klein Tank et al., 2002; Klok and Klein Tank, 2009). Using E-OBS vs. 22, HydroPy was driven by daily temperature and precipitation at 0.1° resolution from 1950–2019. The potential

evapotranspiration (PET) was calculated following the approach proposed by Thornthwaite (1948), including an average day length at a given location. As for HD5–ERA5, the forcing data of surface and sub-surface runoff simulated by HydroPy were first interpolated to the HD model grid and then used to simulate daily discharges.

Investigations by Rivoire et al. (2021) found a higher quality of the ERA5 precipitation over the precipitation in the EOBS data. As a result, we primarily focused on HD5–ERA5 due to its higher quality compared to HD5–EOBS, as analysed in Hagemann and Stacke (2022).

## 3.2 Sea level

### 3.2.1 TRIM–REA6

COSMO–REA6 is the high-resolution regional re-analysis of the German Weather Service (DWD; Bollmeyer et al. (2015)). COSMO–REA6 data were used to force the ocean model TRIM (Tidal Residual and Intertidal Mudflat Model) for the period 1995–2018. The 2D version of TRIM–NP (Kapitza, 2008) is a nested hydrostatic shelf sea model with spatial resolutions increasing from 12.8 km × 12.8 km in the North Atlantic to 1.6 km × 1.6 km in the German Bight. 10-meter height wind components and sea level pressure were used as atmospheric forcing fields. At the lateral boundaries, the astronomical tides from FES2004 atlas (Lyard et al., 2006) were used.

We chose this data set for the main analysis of our work due to the larger region it covers.

### 3.2.2 ECOSMO–coastDat3

The coastDat3 data set is a regional climate reconstruction for the entire European continent, including the Baltic Sea, the North Sea, and parts of the Atlantic (Petrik and Geyer, 2021). The simulation was conducted with the regional climate model COSMO–CLM (CCLM; Rockel et al. (2008)). CoastDat3 covers the period 1948–2019 with a horizontal grid size of 0.11° in rotated coordinates, and the National Centers for Environmental Prediction–National Center for Atmospheric Research (NCEP–NCAR) global reanalysis (Kalnay et al., 1996) was used as forcing and for the application of spectral nudging (von Storch et al., 2000). CoastDat3 data were used to force the physical part of the marine ECOSystem MOdel (ECOSMO) (Schrum and Backhaus, 1999; Daewel and Schrum, 2013) for the period 1948–2019 (Bundesamt für Seeschifffahrt und Hydrographie, 2022). ECOSMO was applied at a spatial resolution of 0.033° longitude and 0.02° latitude, and its domain covers an area from 48.20333° N to 65.90333° N and 4.034667° W to 30.120333° E. The riverine freshwater inflow was taken from a Mesoscale Hydrologic Model streamflow simulation over Europe at 1/16° resolution (Rakovec and Kumar, 2022).

| Data set name | Usage | Variable | Period of available data |
|---|---|---|---|
| HD5–ERA5 | Main analysis<br>Robustness against different parameter settings<br>Robustness against different model-based data sets | discharge | 1979–2018 |
| HD5–EOBS | Time robustness<br>Robustness against different model-based data sets | discharge | 1950–2019 |
| TRIM–REA6 | Main analysis<br>Robustness against different parameter settings<br>Robustness against different model-based data sets | sea level | 1995–2018 |
| ECOSMO–coastDat3 | Time robustness<br>Robustness against different model-based data sets | sea level | 1948–2019 |
| ECOSMO–REA6 | Robustness against different model-based data sets | sea level | 1995–2015 |

**Table 1.** Data set names and their usage in this publication.

### 3.2.3 ECOSMO–REA6

For this data set, the ECOSMO model was forced with COSMO–REA6 data and covers the period from 1995–2015. The initial state was based on a simulation using coastDat2 (Geyer, 2014) forcing from 1990 to 1995. The configuration was otherwise identical to ECOSMO–coastDat3 (Sect. 3.2.2).

While the HD model domain covers the entirety of Europe, the ocean model domains of TRIM and ECOSMO cover only parts of northern Europe. Therefore, our analysis includes a different number of rivers depending on which ocean model was 235 used to generate the sea level data, i.e. either 181 for TRIM-based data or 126 for ECOSMO-based data

### 3.3 Großwetterlagen

Großwetterlagen (GWL) are large-scale weather patterns that form over Europe. Hess and Brezowsky (1969) classified them into 29 different regimes and six circulation types. These weather regimes can persist from a few days up to several weeks in extreme cases. We used a catalogue with this classification system, which was starting back in 1881 and is managed by the 240 DWD. James (2007) stated that there is a strong correlation between the Großwetterlagen and the resulting weather in various regions.

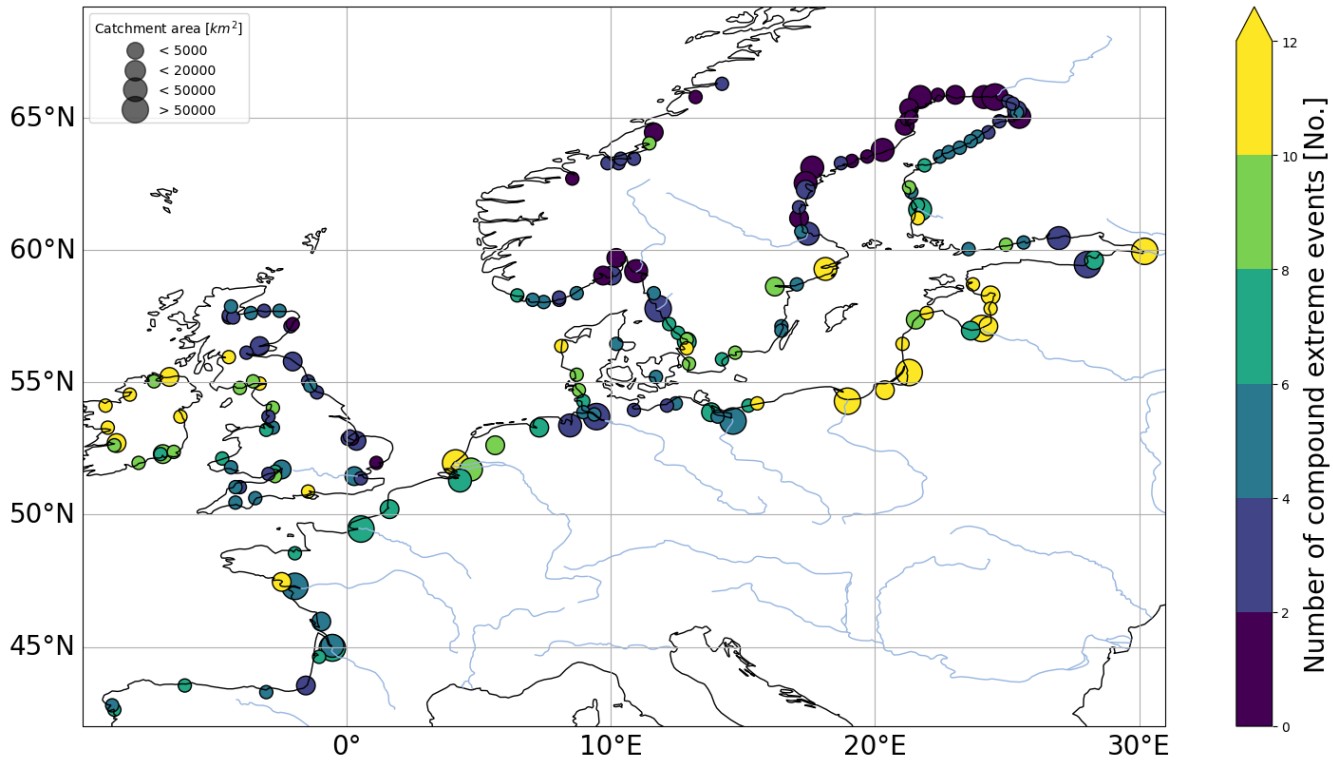

**Figure 2.** Number of compound flood events over a period of 24 years for Northern Europe based on HD5–ERA5 and TRIM–REA6 data. Circle size indicates the catchment size of the corresponding river. The number of discharge and sea level extreme events was limited to two events per year.

## 4   Results

### 4.1   Regional distribution of compound flood events

Fig. 2 shows the distribution of compound flood events for the TRIM–REA6 and HD5–ERA5 data over Northern Europe. A total of 26% of the rivers along the coasts had eight or more compound flood events during the time period 1995–2018.

The regions with the highest number of compound flood events are Ireland and the southeastern Baltic Sea. Furthermore, the west coast of the Baltic states also shows a large amount of compound flood events. The east and south facing coasts of the Bothnian Bay and Bothnian Sea in the Baltic Sea, as well as the eastern British coast, and Skagerrak show the lowest frequency of compound flood events. Similarly, the east coast of Great Britain exhibits a low number of compound flood events, in contrast to the west coast. In general, it can be seen that western facing coasts have a larger number of compound flood events.

Utilising our randomisation method (cf. Sect. 2) yielded Fig. 3 that shows if the amount of observed compound flood events for each river is within or outside the $2\sigma$ interval produced by the randomised data sets. We see that the number of compound

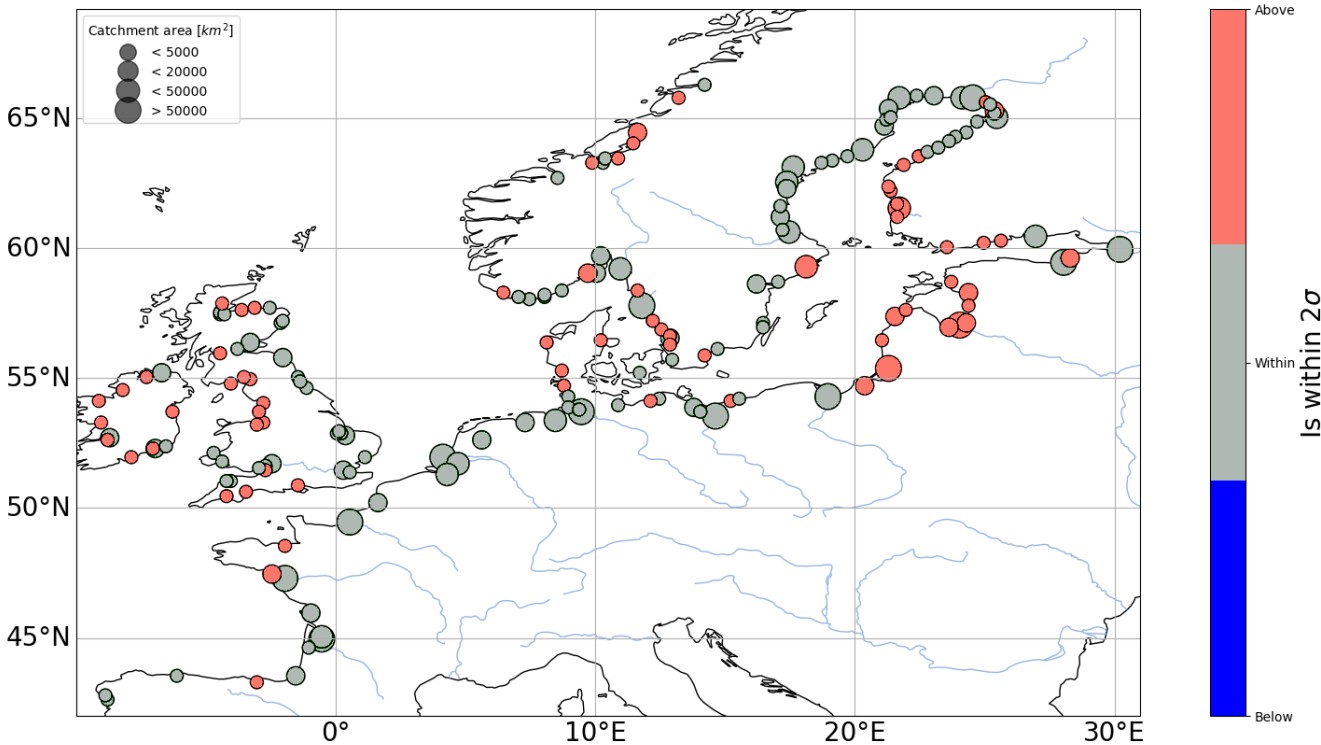

**Figure 3.** Evaluation of compound flood events for rivers in Northern Europe using HD5–ERA5 and TRIM–REA6 data from 1995–2018. The colour indicates if the amount of compound flood events is within (grey), above (red) or below (blue) the expected $2\sigma$ interval. Results are obtained for the winter season with a lag of zero days (cf. Sect. 2).

flood events is outside of the $2\sigma$ interval for the majority of rivers along the westward facing coasts, while the opposite is true for the French west coast.

## 4.2 Robustness of the east-west pattern

To ensure that the pattern seen in Fig. 3 is not the result of sampling effect, parameter, or data choice, we tested different data sets, time periods, and parameters to see whether or not the pattern remains robust. Some images for these tests are in Appendix A for the sake of readability, and they are discussed in the following subsections.

### 4.2.1 Utilisation of various datasets

For the first robustness tests we analysed the combination of ECOSMO–REA6 with HD5–ERA5 (Fig. A1), ECOSMO–coastDat3 with HD5–ERA5 (Fig. 4a), and ECOSMO–coastDat3 with HD5–EOBS (Fig. A2). The pattern remains stable throughout these different data set combinations, with slight variations.

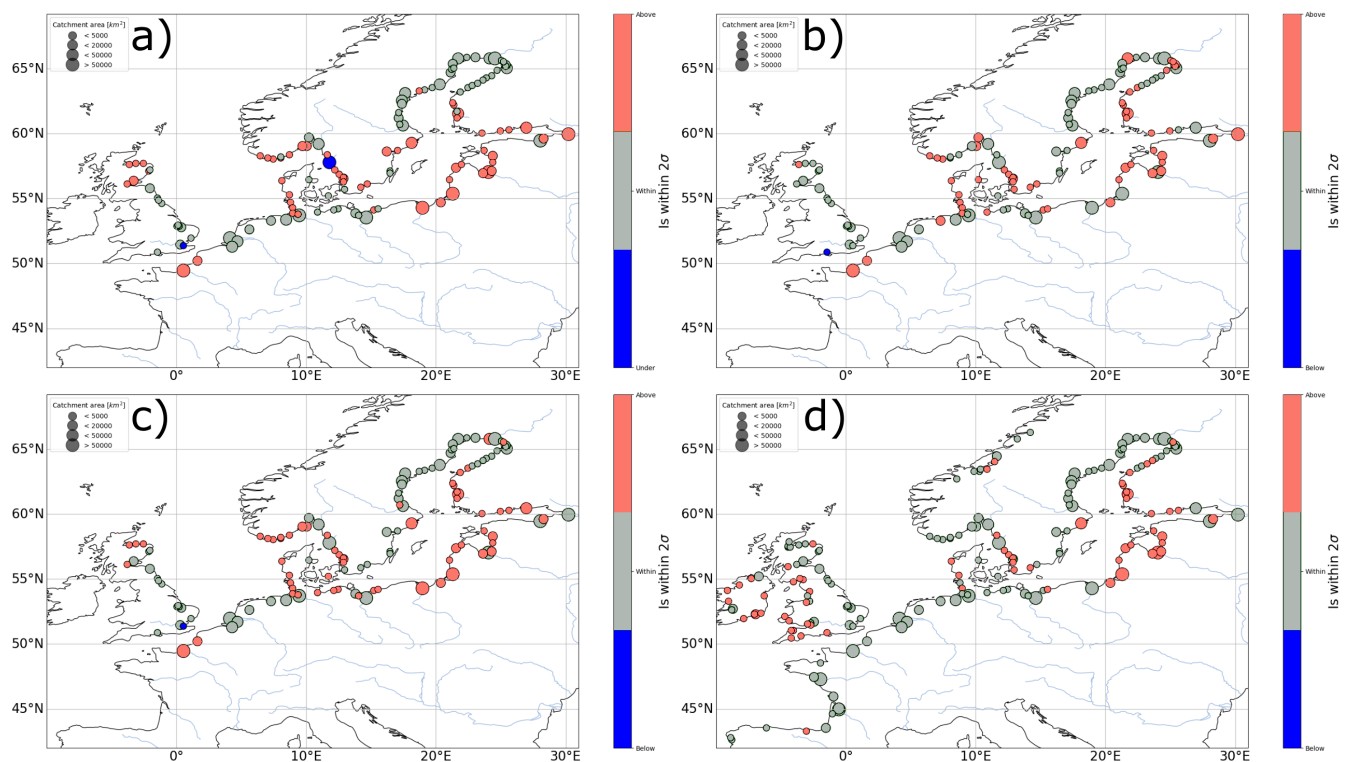

**Figure 4.** Robustness testing. As in Fig. 3 but with different setups. a) Utilised ECOSMO–coastDat3 and HD5–ERA5 data b) ECOSMO–coastDat3 and HD5–EOBS from 1960 to 1989 c) ECOSMO–coastDat3 and HD5–EOBS data from 1990 to 2019 d) TRIM–REA6 and HD5–ERA5 with increased lag from zero to three days.

### 4.2.2 Validation for different time periods

Next, we split the ECOSMO–coastDat3 and HD5–EOBS data into two 30 year sections, from 1960 to 1989 (Fig. 4b) and from 1990 to 2019 (Fig. 4c). The pattern of western facing coasts having a higher number of compound flood events than expected by random sampling remained persistent throughout different time periods, even though it is somewhat more pronounced in the more recent one. This is seen by the generally higher number of rivers above the $2\sigma$ interval, indicating that compound flood events can potentially occur in these months.

Lastly, we added more months to the analysis by adding the month of November (Fig. A4) and finally expanding the time period to last from October to March of the following year (Fig. A5). This resulted in a slightly higher number of rivers being outside of the $2\sigma$ interval.

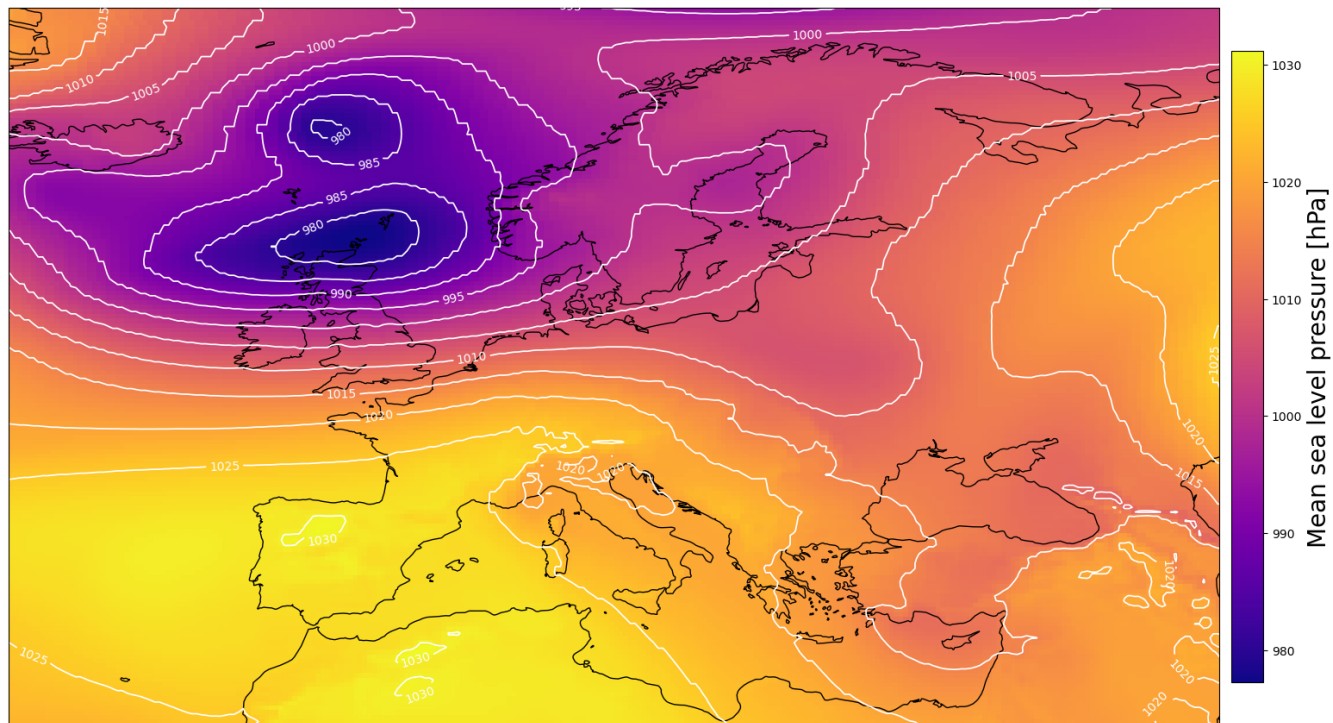

**Figure 5.** Map of the daily mean atmospheric pressure over Europe on the 08 December 2011. The characteristic low pressure centre of the Großwetterlage Cyclonic Westerly is located north of Scotland (ERA5, 2022).

### 4.2.3 Changes to parameters and randomisation

As a first test, we changed the lag from zero to three days which is shown in Fig. 4d). This resulted in a slightly higher number of river catchments within the expected interval. Furthermore, we also tested the second randomisation method described in Sect. 2 in order to interrupt possible dependencies. For this, we randomised the order in which the years appear in our sea level data sets. The biggest difference with this simpler randomisation approach was that two additional rivers on the British east coast are below the $2\sigma$ deviation.

Additionally, we compared the influence of two different thresholding methods on the results, namely self-tuning thresholds (Fig. 3) and plain percentiles (Fig. A3), both described in Sect. 2. Both methods lead to nearly identical results.

### 4.3 A common meteorological driver for compound flood events

To see if the regions with a higher than expected number of compound flood events have a common large-scale meteorological driver we analysed the meteorological situation during these events. The coordinates of those regions are available in Table 2.

For our analysis, we focused first on the German–Danish west coast. This coast contains the five rivers Stora, Ribe A, Bongsieler Kanal, Eider, and Oste. Our goal was to scrutinise whether large-scale compound flood events in these rivers have

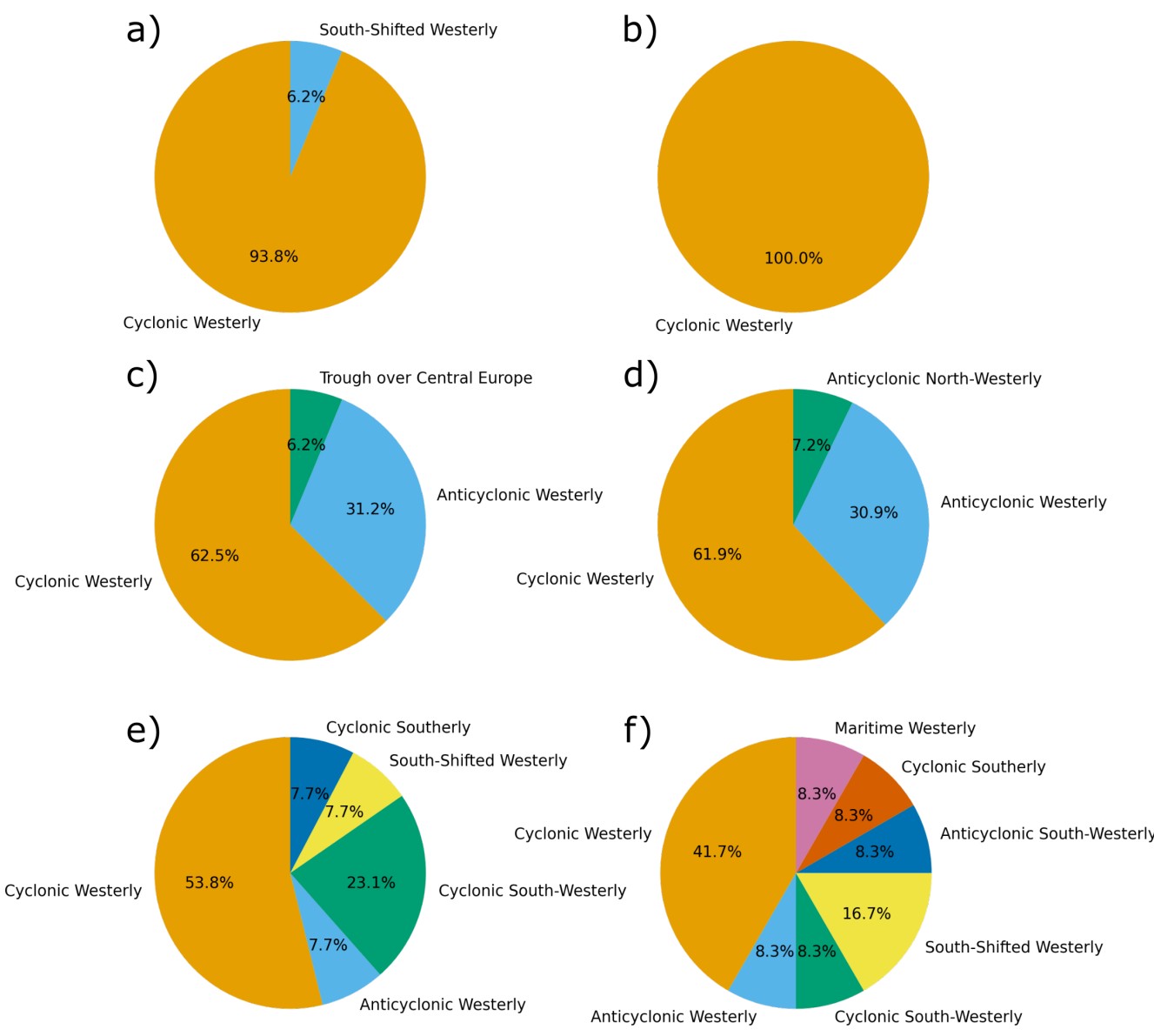

**Figure 6.** Distribution of Großwetterlagen that occurred during compound flood events in Europe. The following regions were analysed: a) German-Danish west coast b) Western facing coast of Sweden. c) Western facing coast in the Bothnian Sea d) West coast of the Baltic states e) West coast of Great Britain f) West coast of Ireland. Coordinates of those regions are given in Table 2.

| Region | Coordinates |
|---|---|
| West coast of the Baltic states | 54.52° N–59.00° N × 20.00° E–24.80° E |
| West coast of Great Britain | 50.79° N–55.99° N × 4.85° W–2.50° W |
| German-Danish west coast | 53.81° N–56.46° N × 8.02° E–9.12° E |
| Western facing coast in the Bothnian Sea | 61.12° N–62.46° N × 21.18° E–21.80° E |
| West coast of Ireland | 52.48° N–54.72° N × 9.30° W–7.90° W |
| Western facing coast of Sweden | 55.37° N–59.37° N × 10.90° E–13.20° E |

**Table 2.** Regions and their corresponding coordinates sorted in alphabetical order. They are used for the analysis in Sect. 4.3. These regions are also utilised in the visualisation of the results in Fig. 6.

a specific Großwetterlage as their common meteorological driver. For this, we decided to examine which Großwetterlage is present when at least four of the five rivers have a compound flood event simultaneously. This requirement resulted in 16 separate compound flood events based on ECOSMO–coastDat3 + HD5–ERA5 and ECOSMO–coastDat3 + HD5–EOBS data. 15 of these events appeared during the Großwetterlage Cyclonic Westerly (Fig. 5), with only one appearing during
Cyclonic North-Westerly (Fig. 6a). The Großwetterlage Cyclonic Westerly is associated with strong westerly winds and higher than normal precipitation (Gerstengarbe et al., 1999) that can cause storm surges and river floods, respectively, which in combination can lead to compound flood events. Also, our analysis showed that at least 75% of the compound flood events for each river along the German–Danish west coast happened during this specific Großwetterlage. This made it the predominant Großwetterlage during compound flood events in this area.

Similar results were found for the Swedish west coast in Kattegat and Skagerrak. There, all 7 events that involved at least 4 rivers appeared during the Großwetterlage Cyclonic Westerly, based on ECOSMO–coastDat3 and HD5–ERA5 data (Fig. 6b).

In the western facing coast of the Bothnian Sea, Cyclonic Westerly remained the predominant Großwetterlage. About two-thirds of the events occurred during Cyclonic Westerly and one-third during Anticyclonic Westerly (Fig. 6c). In the coastal area of the Baltic states, we observed again a distribution of roughly two-thirds of the events appearing during Cyclonic Westerly
and one-third during Anticyclonic Westerly (Fig. 6d). Anticyclonic Westerly is known to lead to precipitation in the area of the Baltic countries (Jaagus et al., 2010), which in combination with the south-eastern wind direction are responsible for around a third of the flood compound events in the Baltic and western facing Finnish area, due to the orientation of their coastline. For the western facing coast of Great Britain, we found that half of the compound flood events happen during Cyclonic Westerly, a quarter of the events during Cyclonic South-Westerly and the remaining during other Großwetterlagen (Fig. 6e).

Unlike the other cases, we didn't observe any predominant Großwetterlage for compound events in Ireland, with Cyclonic Westerly accounting for less than half of the observed Großwetterlagen during compound flood events (Fig. 6f).

Furthermore, we investigated possible correlations between the duration of a Großwetterlage and the occurrence of compound flood events. We found that compound flood events can occur during short Großwetterlagen that only last 3 days, which is by definition the minimum duration, as well as Großwetterlagen that remain over several weeks. Therefore, we didn't find
any direct correlation. Additionally, we didn't observe any specific sequence of Großwetterlagen that leads to an increased risk

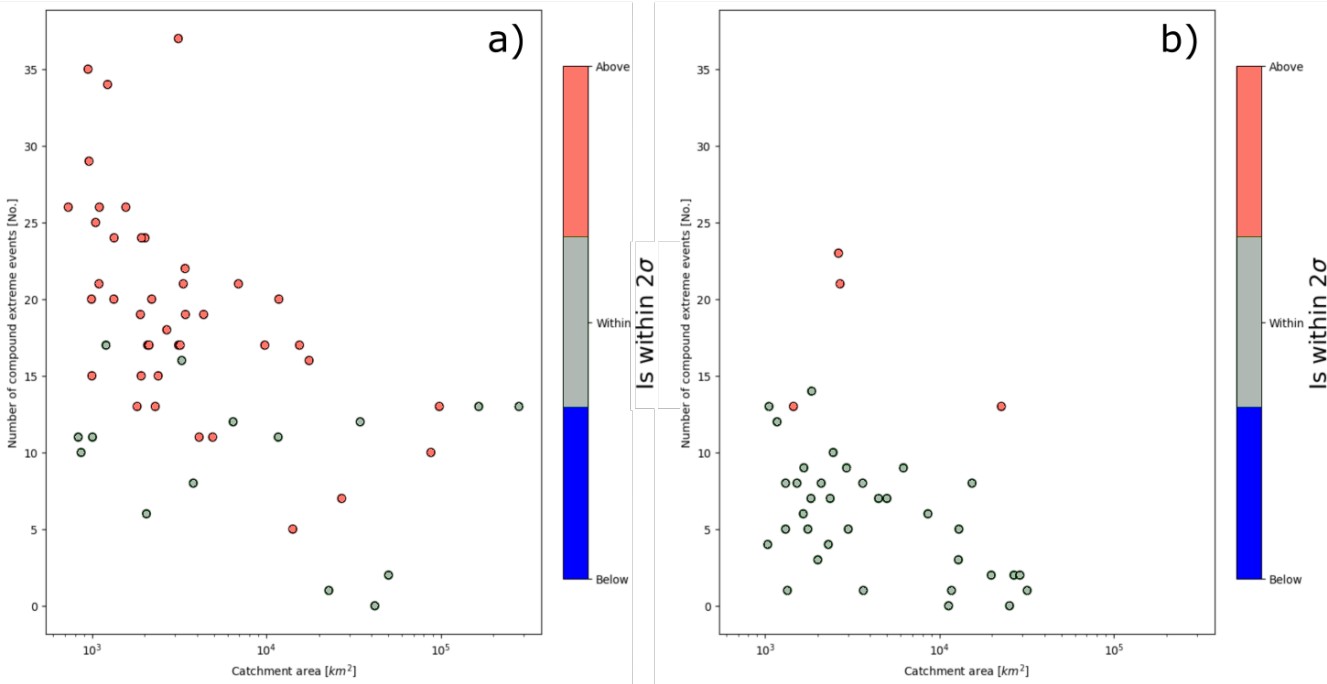

**Figure 7.** Number of extreme events for North Europe over a period of 24 years plotted over the river's corresponding catchment area for HD5–ERA5 and TRIM–REA6 data using percentiles. The colour displays if the amount of observed compound flood events is within the expected $2\sigma$ deviation. Contains only rivers that are either on the a) western or b) eastern facing coasts.

of compound flood events. Finally, the kind of Großwetterlage, which follows or precedes the Großwetterlage that causes a compound flood event, seems to be random.

## 4.4    Correlation between the number of compound flood events and catchment size

We analysed if there is any connection between the catchment size and the frequency of compound flood events. For this, we
plotted the number of compound flood events against the size of the catchment area of each river (Fig. 7). The catchment size of each river was obtained from the HD model grid. The analysis was done separately for rivers based on their orientation along the coasts. Furthermore, the rivers were coloured red if the number of compound flood events is above the $2\sigma$ interval of randomised sea level data, blue if below the interval, and grey otherwise, such as in Fig. 3. It can be seen that there is a clear correlation between the cardinal direction of the estuary and the number of compound flood events either being inside
or outside of the $2\sigma$ interval. The western facing coasts (Fig. 7a) were mostly above the $2\sigma$ interval and showed generally a higher number of compound flood events. Contrarily, the eastern facing coasts (Fig. 7b) exhibited a lower amount of compound flood events and are mostly within the expected margin. Additionally, it can be seen that the number of compound flood events declined with increasing catchment area, regardless of cardinal direction.

## 5 Discussion & Conclusions

In the present study, we conducted a coherent spatial analysis on the dependence of storm surges and discharge extreme events as drivers of compound flood events over Northern Europe. For this analysis, we introduced a method to analyse compound events by randomising one of the data sets to generate independent data. To our knowledge, this is the first study on compound flood events over entire Europe that does not utilise copulas. As mentioned in the introduction, copulas add unknown amounts of uncertainty to the analysis. Our method on the other hand is easy to implement and the uncertainty is given by the standard deviation. One limitation of this method is that it cannot quantify the dependence between discharge and sea level.

Using different datasets of daily discharge and sea level, we detected a distinct pattern of westward-facing coasts having a higher number of compound flood events than expected by chance (Fig. 3, Fig. 7). These coasts were located in the European storm-track corridor comprising the British Isles, northern Germany, Denmark, and southern Sweden (Feser et al., 2015). Due to the mostly prevailing western winds, the rivers on the eastern coasts showed a lower number of compound flood events, which are usually within the expected range of two standard deviations. This finding is consistent with the results of Paprotny et al. (2018b), who noted a strong dependency in their rank correlation for west-facing coasts in Northern Europe. Khanal et al. (2019) and Kew et al. (2013) likewise reported that the most extreme events in the Rhine delta are connected to westerly winds. Similarly, (Svensson and Jones, 2004) reported a strong dependence between discharge and storm surge events for western Great Britain. We identified the Großwetterlage Cyclonic Westerly as the common meteorological driver for the occurrence of large-scale compound flood events in North and Baltic Sea regions.

In parts of the Baltic and western facing Finnish coasts, the Großwetterlage Anticyclonic Westerly additionally contributed to the generation of compound flood events (about one-third). For Ireland, a distinct Großwetterlage could not be identified as a driver of compound flood events. We speculate that this might be because it offers a wide angle of attack for storm surges.

Additionally, we were able to demonstrate that the detected spatial distribution remains stable for various sources of uncertainty. Our results proved to be robust against the utilisation of different forcing data for the simulation of discharge and sea level data, parameter settings, and randomisation approaches. Furthermore, the pattern remained relatively stable despite the ongoing climate change since the 1960s. There was a certain amount of variation in the pattern, which can be attributed to randomness and the different setups. Due to the limited number of compound flood events, even small variations to their definition, like changes in the allowed lag, have a minor influence on the results. In all cases, the pattern was present, even though it was sometimes more or less pronounced.

In addition, we demonstrated that there exists a correlation between river catchment size and the number of compound flood events. It can be seen that, regardless of the estuary orientation, the number of compound flood events declined with increasing catchment size. The reason for this might be that rivers with smaller catchment areas are capable of reacting faster to precipitation that appears during the storm events, which also causes the storm surges. There is some variation in the distribution, as expected by the design of the test, which resulted in around 5% of the data points being labelled incorrectly.

Our analysis here is associated with some caveats that have to be considered. We note that the utilisation of the $2\sigma$ interval in our analysis comprises some amount of uncertainty. As a result, it can be expected that 5–9 rivers will be incorrectly labelled,

based on the size of the data set. Another problem for our analysis was the very short time frame that was accessible with the TRIM–REA6 and ECOSMO–REA6 data of 24 and 21 years respectively. Furthermore, it is possible that the model based data sets contain systematic errors. Despite the detected pattern being robust, it is possible that the absolute number of compound flood events may deviate from the actual number. Furthermore, the de-clustering time of 4 days might be too short for some of the longest rivers that may contain very long extreme events. The lack of a parametric model impedes the possibility of deriving engineering quantities for design events to test flood protection structures.

Future work can further examine these findings by using ensemble data over a longer time frame, e.g. 50 years and more. This could enable generating a distribution for the number of compound flood events, based on the compound flood events detected in the individual ensemble members. As a result, it would become possible to calculate how many compound flood events to expect on average in each river. This reduces the influence of randomness, by not having to rely on the compound flood event number detected in a single data set. Additionally, future studies could focus on locations in close spatial proximity along the western facing coasts for which long time series of daily sea level and discharge data are available. Another interesting question, which needs further investigation, is why the vast majority of compound flood events on the west coasts happen during Cyclonic Westerly, while not every one of these Großwetterlagen results in compound flood events. Understanding what makes them different might offer opportunities to identify them early and set contingency plans into motion.

In order to support future risk assessments, it will be important to analyse how compound events will change under different climate scenarios and sea level rises (Zscheischler et al., 2018). First, the frequency change of general flood events with respect to the current standards for extreme events might change especially with increasing sea levels. Second, it will be interesting to analyse, if our observed pattern caused by the Großwetterlage remains similar, or if we will see changes to it due to, e.g changes in the occurrence rate of this specific Großwetterlage. This is important since it is well known that there have been frequency changes in the past as reported by Grabau (1987) and Dietz (2019). Hoy et al. (2013) found that the frequency of Cyclonic Westerly was declining during the first half of the last century, before strongly rising between 1970 and 2000. This leads to the question of how the frequency of compound flood events might change for entire Europe. Third, the vast majority of compound flood events are currently centred around the winter season. It is important for our general understanding to investigate, if the seasonal distribution itself will change, maybe with more events in summer, or if the distribution stays the same with different numbers.

*Data availability.* The HD discharge data were published as Hagemann and Stacke (2021) and can be obtained at the World Data Centre for Climate (WDCC) of the German Climate Computing Centre (DKRZ). TRIM and ECOSMO sea level data will be made available by the authors, without undue reservation, to any qualified researcher upon request. ERA5 reanalysis data are available at the Climate Data Store: https://cds.climate.copernicus.eu/. DWD Großwetterlagen are available at https://www.dwd.de/DE/leistungen/grosswetterlage/grosswetterlage.html

# Appendix A: Appendix: Images for parameter changes

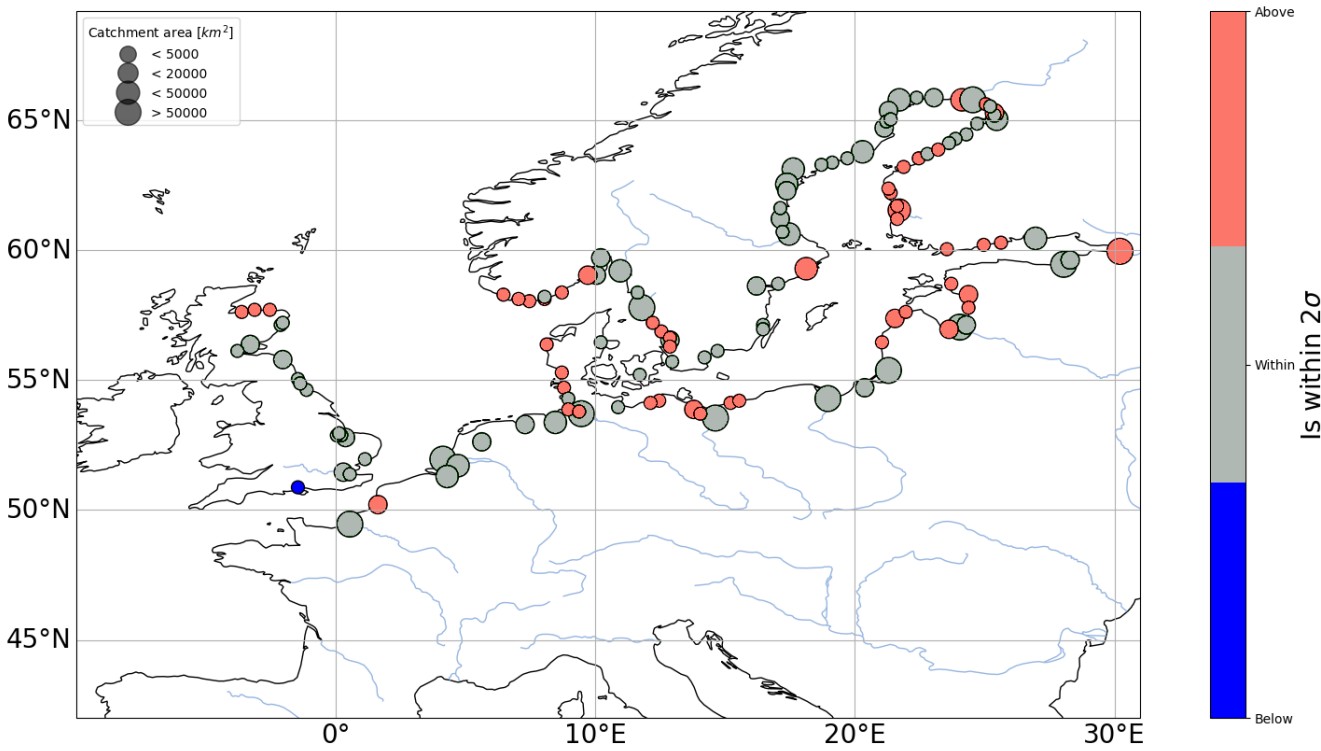

**Figure A1.** HD5–ERA5 as in Fig. 3 but with ECOSMO–REA6 for the sea level data.

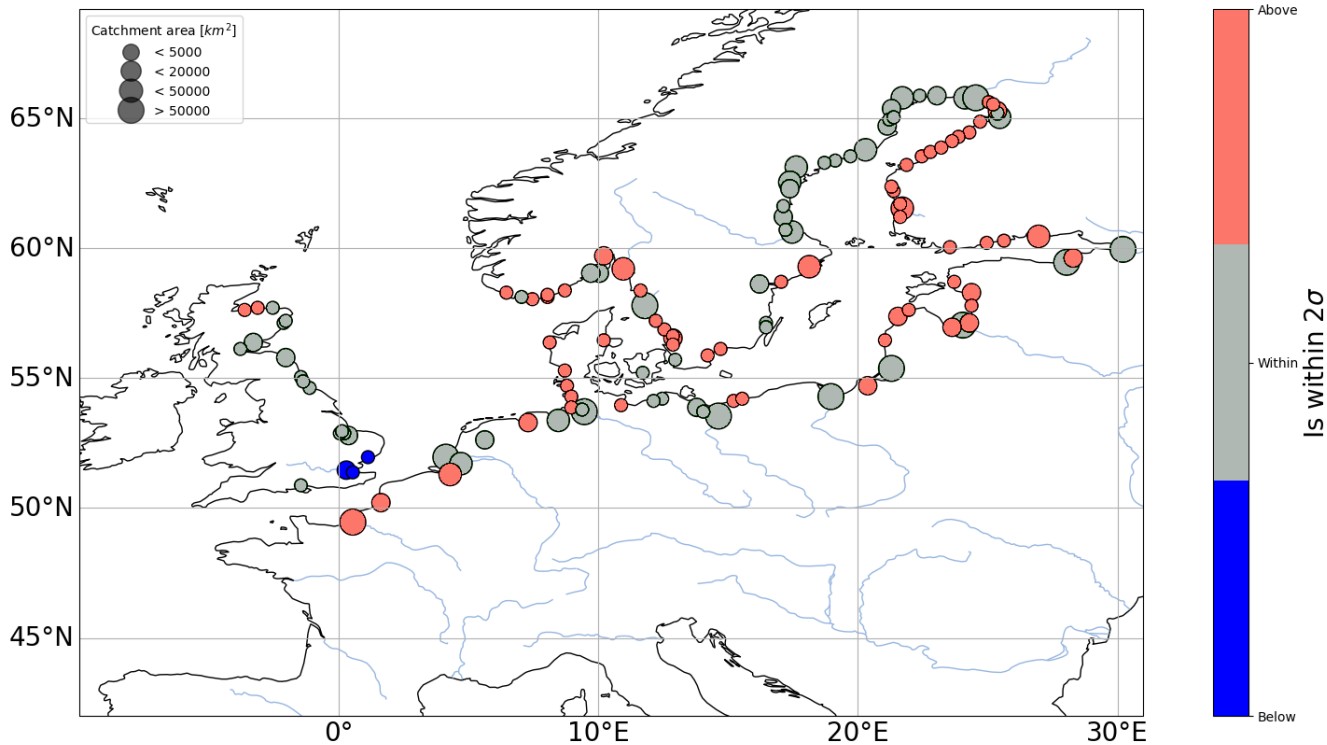

**Figure A2.** As in Fig. 3 but with ECOSMO–coastDat3 and HD5–EOBS data.

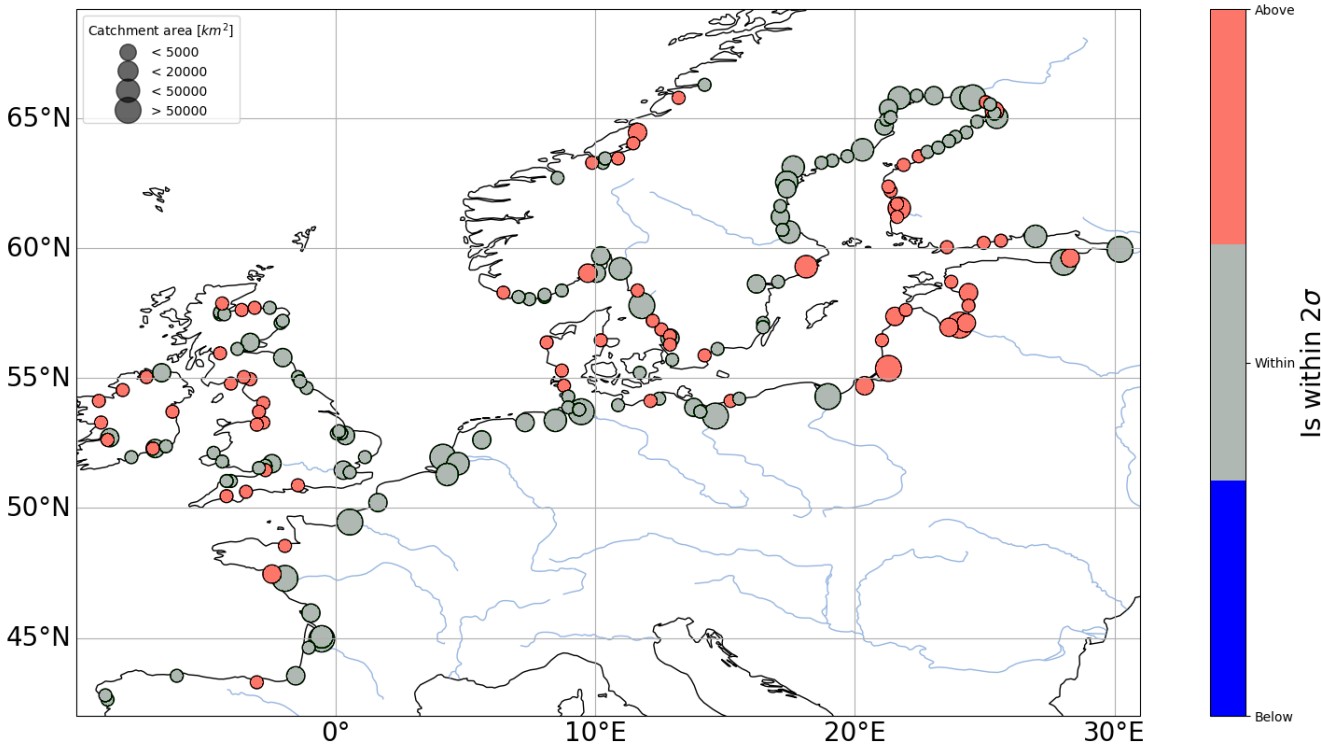

**Figure A3.** TRIM–REA6 and HD5–ERA5 as in Fig. 3 but utilising normal percentile instead of the adaptive thresholds.

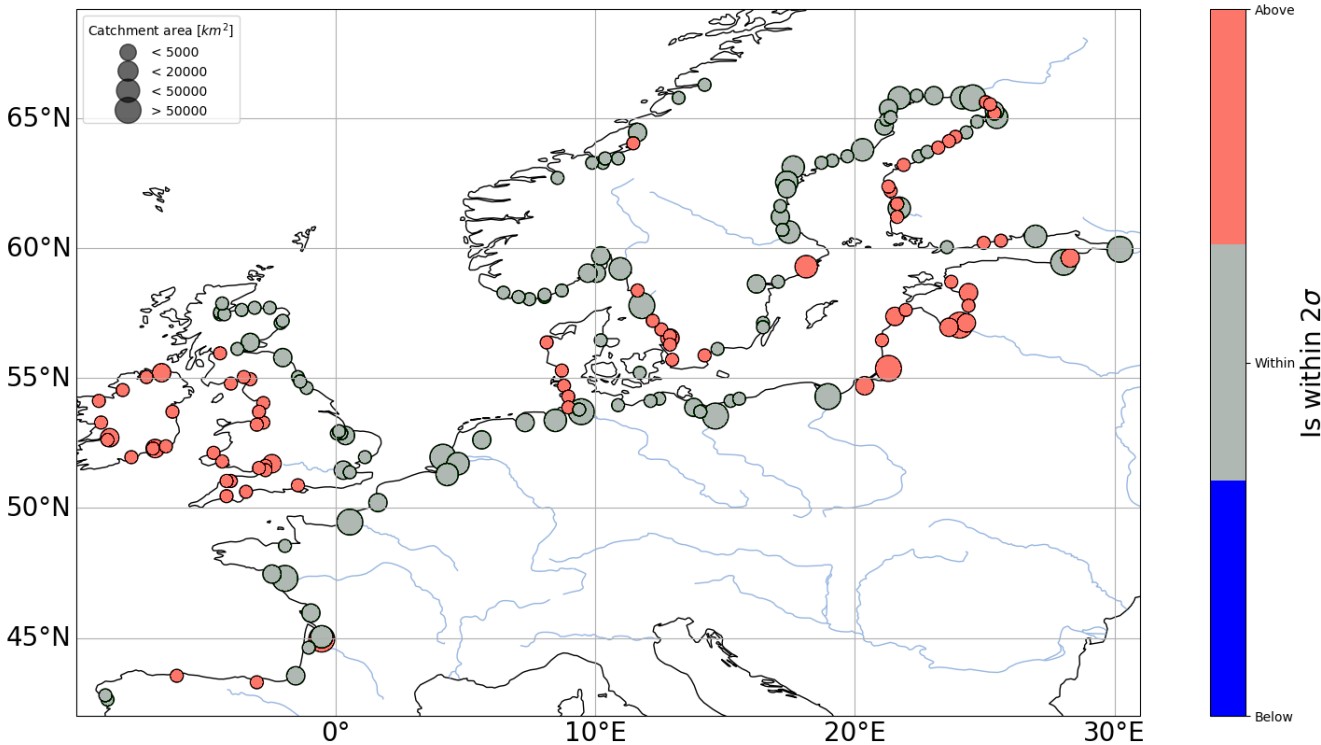

**Figure A4.** TRIM–REA6 and HD5–ERA5 as in Fig. 3 but for the the the months November to February.

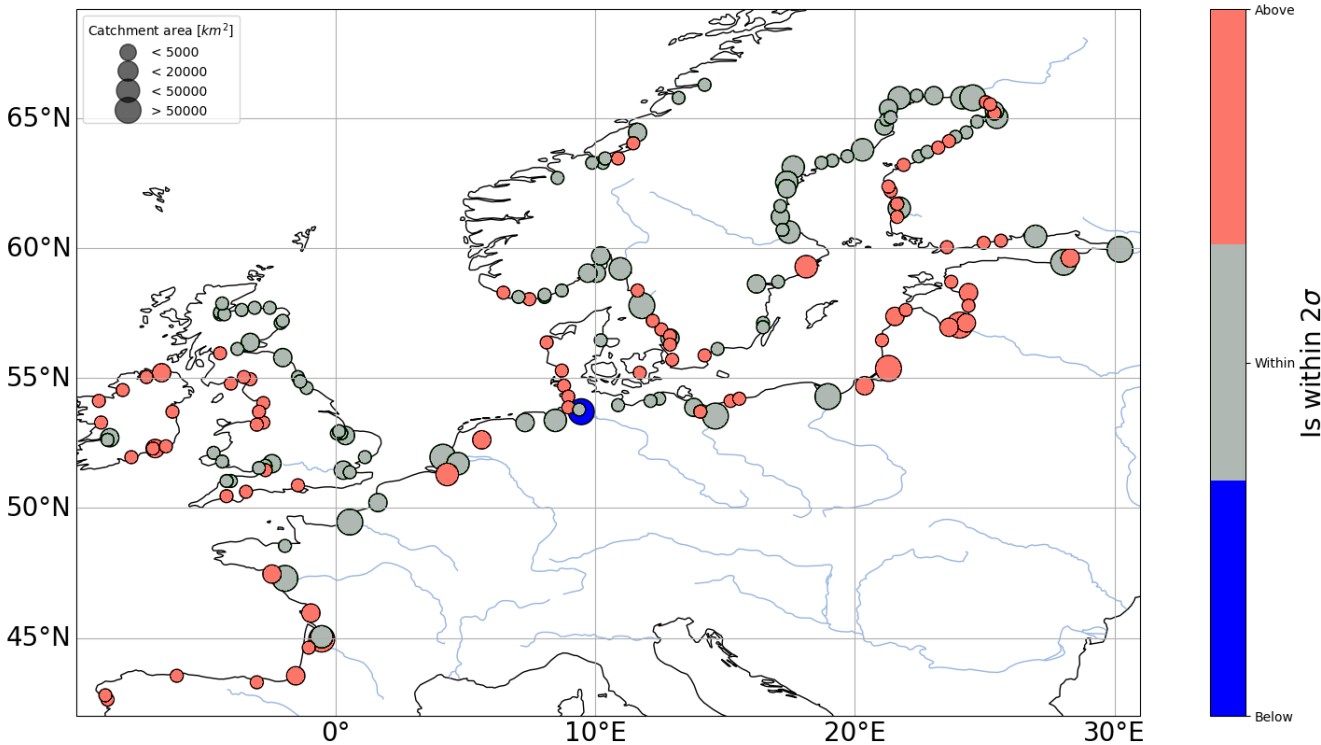

**Figure A5.** TRIM–REA6 and HD5–ERA5 as in Fig. 3 but for the months October to March.

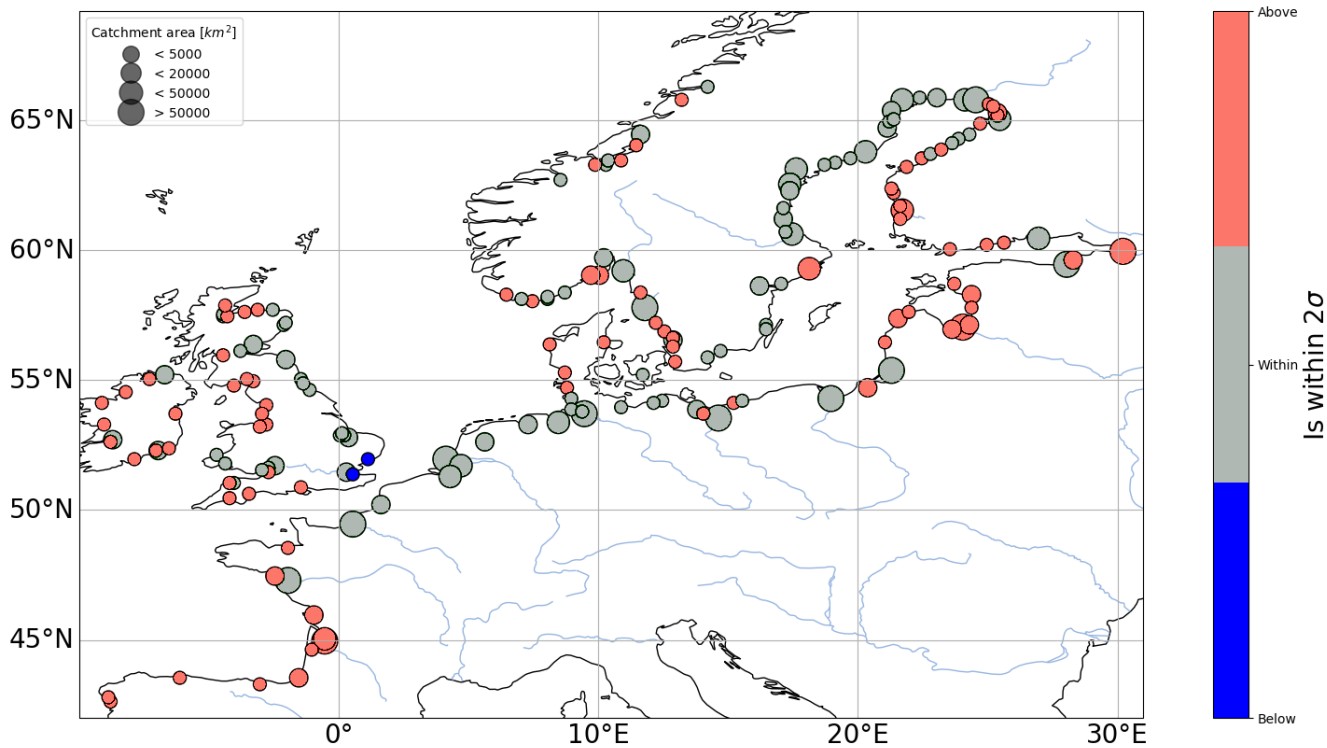

**Figure A6.** TRIM–REA6 and HD5–ERA5 as in Fig. 3 but with swapping the years for randomisation instead of the method described in Sect. 2.

*Author contributions.* PH developed the analysis methods, performed the data analysis and wrote the manuscript. SH initiated the study and contributed the HD discharge data while LG and UD generated the TRIM and ECOSMO sea level data, respectively. CS revised the manuscript. RW and SH revised the manuscript and contributed to the interpretation of the results. RW acquired the funding and SH, RW, and CS supervised the research activities.

*Competing interests.* The authors declare that they have no known competing financial interests or personal relationships that could have
395 appeared to influence the work reported in this paper.

*Acknowledgements.* This research was financed with funding provided by the German Federal Ministry of Education and Research (BMBF; Förderkennzeichen 01LR2003A). Furthermore, this research is a contribution to the PoFIV program of the Helmholtz Association.

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
