# Peer review of "Compound Flood Events: Analysing the joint occurrence of extreme river discharge events and storm surges in Northern and Central Europe"

_Natural Hazards and Earth System Sciences, 2022_

## Author Response (AR1)

**Reply to reviewers' comments**

We thank the two reviewers for their thorough reading of the manuscript and their valuable remarks that helped us to improve the manuscript. In the following, the original reviewer comments are given in italic and all line numbers and figure numbers refer to the original submitted version that was reviewed if not mentioned otherwise.

**Reply to review of reviewer 1**

*The study gives an overview of past analyses of compound events in Europe, and proposes an empirical approach not relying on copulas to build a new climatology of compound discharge/surge peaks of NW European rivers. It concludes that westward facing coasts show more than random CEs, while eastward facing coasts are not shown to demonstrate an expected nr of events. For most areas Cyclonic Westerlies are held responsible. Large basins show fewer CEs.*

*The method is robust – as demonstrated by using different datasets and system parameters – and yields results that are physically understandable: the westerlies, certainly when their directional wind fields constrain the fetch over which surges can develop, generate both a surge and anomalous rainfall, which for smaller catchments comes to joint water level peaks. This expected physical interpretation is both a demonstration of skill of the method, but also a bit disappointing because it is not really a surprising result.*

We thank reviewer 1 for the positive evaluation of our manuscript.

*Other studies have elaborated on the physical backgrounds of CEs in westerly storm track climates, and extended it further by also including explicit considerations of time lag (Kew et al 2013) and hydrological/meteorological memory (Khanal et al, 2019).*

We added the following sentence to line 318:

Khanal et al. (2019) and Kew et al. (2013) likewise reported that the most extreme events in the Rhine delta are connected to westerly winds.

*It is not really clear how the findings in this paper will contribute to practical applications, such as enhanced predictability, statistical underpinning of infrastructure design and others. This is not a requirement for a study like this to be published, but some extension of the implications of these findings for the better understanding or development of societal applications would be welcome.*

We believe that it is at this stage too early to speculate about possible societal applications. As discussed in lines 341-349, more scientific work is needed to gather a better understanding on what makes CW that cause CFE different from those that do not. Our work should therefore rather be seen as a pointer in a direction for further investigation.

*       Other definitions of compound events than the SREX report are provided by Leonard et al (2014) and (already cited) Zscheischler et al (2018).*

We added the following text to line 28:

A more general definition was proposed by Leonard et al. (2014), who defined it as "an extreme impact that depends on multiple statistically dependent variables or events."

\*      *Relevance to consider compound events in risk assessment are also clearly exposed by de Ruiter et al (2020)*

We added the citation to line 21:

The occurrence of extreme flood events either simultaneously or in close succession can lead to severe damage, which greatly exceeds the damage those events would cause separately (de Ruiter et al., 2020, Xue et al., 2022).

\*      *The color scale in Figure 1 is not very intuitively ranging between low to high values*

Changed colormap of Figure 1 to viridis:

[Figure]

\*      *Line 290: a Western Cyclone GWL can not last for weeks, I would say. As shown by Kew, series of low pressure systems may well affect compound events in large river basins like the Rhine, provided a time lag is allowed*

As shown in Table 4 on page 21 of the Großwetterlagen catalogue v.7 (https://www.pik-potsdam.de/en/output/publications/pikreports/.files/pr119.pdf), it is possible for CW to last for over 3 weeks.

We changed line 222 to:

These weather regimes can persist from a few days up to a few weeks in extreme cases.

We changed line 290 to:

We found that compound flood events can occur during short Großwetterlagen that only last 3 days, which is by definition the minimum duration, as well as Großwetterlagen that remain over two weeks.

List of added references:

Kew, S., Selten, F., Lenderink, G., and Hazeleger, W.: The simultaneous occurrence of surge and discharge extremes for the Rhine delta, Natural Hazards and Earth System Sciences, 13, 2017–2029, https://doi.org/10.5194/nhess-13-2017-2013, 2013.

Khanal, S., Lutz, A. F., Immerzeel, W. W., Vries, H. d., Wanders, N., and Hurk, B. v. d.: The impact of meteorological and hydrological memory on compound peak flows in the Rhine river basin, Atmosphere, 10, 171, https://doi.org/10.3390/atmos10040171, 2019

Leonard, M., Westra, S., Phatak, A., Lambert, M., van den Hurk, B., McInnes, K., Risbey, J., Schuster, S., Jakob, D., and Stafford-Smith, M.: A compound event framework for understanding extreme impacts, Wiley Interdisciplinary Reviews: Climate Change, 5, 113–128, https://doi.org/10.1002/wcc.252, 2014

Ruiter, Marleen C. de, Anaïs Couasnon, Marc J.C. van den Homberg, James E. Daniell, Joel C. Gill, and Philip J. Ward. 2020. "Why We Can No Longer Ignore Consecutive Disasters." Earth's Future. John Wiley and Sons Inc. https://doi.org/10.1029/2019EF001425.

Xu, H., Tian, Z., Sun, L., Ye, Q., Ragno, E., Bricker, J., Mao, G., Tan, J., Wang, J., Ke, Q., et al.: Compound flood impact of water level and rainfall during tropical cyclone periods in a coastal city: the case of Shanghai, Natural Hazards and Earth System Sciences, 22, 2347–2358, https://doi.org/10.5194/nhess-22-2347-2022, 2022

**Reply to reviewers' comments**

We thank the two reviewers for their thorough reading of the manuscript and their valuable remarks that helped us to improve the manuscript. In the following, the original reviewer comments are given in italic and all line numbers and figure numbers refer to the original submitted version that was reviewed if not mentioned otherwise.

**Reply to review of reviewer 2**

*The authors contend that copulas add unnecessary uncertainty to the estimation of compound event frequency. They therefore apply a non-parametric randomization test to find the spatial distribution of compound high river discharge -storm surge events across Northern Europe. Similar randomization techniques have been applied in past e.g., Svensson and Jones (2002) and Zhang et al. (2013), but the scale of this analysis arguably provides some novity. The compound events that impact most of the rivers on the German-Danish west coast are shown to be linked by common metrological drivers as characterized by weather types defined by the German Weather Service. In my opinion, the preprint has the bones of a good paper but more work is required before it is worthy of publication in Natural Hazards and Earth System Science Journal.*

We thank reviewer 2 for the overall positive evaluation of our manuscript and very detailed comments.

*The introduction contains a wealth of relevant information; however, it does not flow particularly well. I believe a major cause of this is the poor internal structure of certain paragraphs. The first sentence should summarize the main point being in the rest of the paragraph. The paragraph starting on line 31 is a prime example where this is not the case. It starts by stating "Several studies conducted over the last years have shown the importance and catastrophic nature of compound flood events for several locations" but no catastrophic compound flood events are subsequently discussed. Instead studies that explore the correlation between the flooding drivers are listed. The paragraph ends describing results from Hendry et al. (2019), a relevant reference, since this study goes on to investigate potential correlations between a river's catchment size and the number of compound flood events, but out of place here.*

We changed the language at several places to avoid a mix of British and American English (e.g., changed randomized to randomised). In addition, we went through the manuscript and corrected a few typos as well as revised a few sentences for better readability.

The following changes were made to the introduction, containing also changes based on the specific comments:

[revised manuscript text omitted]

*Finally, paragraphs should never start with linking words such as "therefore" or "Next".*

The following paragraphs were changed:
       Line 71: In the present study, we analyse compound flood events by focusing ….

       Line 247: To test the robustness for different time periods, we split the ECOSMO-coastDat3 and HD5-EOBS data into two 30 year sections, from 1960 to 1989 (Fig. 3b) and from 1990 to 2019 (Fig. 3c).

*The term "potential compound flood events" needs to be defined. It appears the term "potential" is adopted in other studies as they only concentrate on the drivers of flooding and not the pathway or receptors component of the flooding process.*

The idea was with this specific phrasing to take into account that the identified events might not be considered "extreme" due to local properties like topography and flood protection. We removed "potential" since the literature generally refers to them as compound flood events.
Additionally, we removed the sentence starting from line 28:
In the following text we will note "potential compound flood events" as "compound flood events" […].

*The introduction (L62-64) contains some very strong statements regarding the suitability of copulas in climate research. These statements must be either backed up with references or toned down (see specific comments).*

We added two additional references to line 62 and applied the suggested change from the specific comments:

However, copulas introduce an additional amount of uncertainty (Heffernan and Tawn, 2004) and may require a large number of data points for robust tail dependence calculation (Moran, 1957).

Furthermore, we toned down the statement in lines 63-64:

In climate research, the amount of available data points is often too small for this kind of analysis with many studies operating at merely 30 extreme events.

Please note that we provided further statements on the underlying assumptions like dependence (Frahm et al., 2005) and results (Serinaldi et al., 2015) already in the original text.

*The discussion of the limitation of the method could be further developed. For example, imposing a separation criterion of four days means events may not be independent in some of Europe's longest rivers (I agree this problem is not unique to this study). Another limitation is that not fitting a parametric model prevents the estimation of useful engineering quantities such as design events.*

We added the following lines to line 340:

Furthermore, the de-clustering time of 4 days might be too short for some of the longest rivers that may contain very long extreme events. The lack of a parametric model impedes the possibility of deriving engineering quantities for design events to test flood protection structures.

*Certain Figures can be improved. The maps look distorted and not all rivers are shown.*

It is correct that not all rivers are shown with their river network. Showing all rivers in a detailed way at this zoom level would make the figures harder to read. The relevant location are the rivers in the figures indicated by the large circles. We considered only rivers where discharge observation were available for the evaluation of the HD model discharge (see Hagemann and Stacke 2022)

*Any locations including rivers, seas and oceans mentioned in the text need to be identified in the Figures.*

As suggested, we added the following image to the end of the "Methods" section.

[Figure]

**Figure 1.** This figure contains the catchments, regions, and seas that are mentioned by name throughout the study. The first five entries in the colorbar contain maritime zones with highlighted catchment areas of rivers that discharge into them. The last five entries show the catchment area of five rivers on the German-Danish western coast.

We added the following after Line 157:

The domains of all catchments, regions, and seas that we mention by name for various reasons in this study can be seen in Fig.1.

The following lines were changed:

Line 230: The east and south facing coasts of the Bothnian Bay and Bothnian Sea in the Baltic Sea, as well as the eastern British coast, and Skagerrak show the lowest frequency of compound flood events.

Line 278: In the western facing coast of the Bothnian Sea, Cyclonic Westerly remained the predominant Großwetterlage.

*A figure showing the proportion of each weather type responsible for the compound events at each location either as mini pie or bar charts or Figure 7 in Camus et al. (2022) would add value. This would provide evidence for statements such as "For Ireland, a distinct Großwetterlage could not be identified as a driver of compound flood events. We speculate that this might be because it offers a wide angle of attack for storm surges."*

We added after Line 264:

The coordinates of those regions are available in Table 2.

| Region | Coordinates |
| --- | --- |
| West coast of the Baltic states | 54.52° N–59.00° N × 20.00° E–24.80° E |
| West coast of Great Britain | 50.79° N–55.99° N × 4.85° W–2.50° W |
| German-Danish west coast | 53.81° N–56.46° N × 8.02° E–9.12° E |
| Western facing coast in the Bothnian Sea | 61.12° N–62.46° N × 21.18° E–21.80° E |
| West coast of Ireland | 52.48° N–54.72° N × 9.30° W–7.90° W |
| Western facing coast of Sweden | 55.37° N–59.37° N × 10.90° E–13.20° E |

**Table 2.** Regions and their corresponding coordinates sorted in alphabetical order. They are used for the analysis in Sect. 4.3. These regions are also utilised in the visualisation of the results in Fig. 6.

Furthermore, we added the following image as requested:

[Figure]

**Figure 6.** Distribution of Großwetterlagen that occurred during compound flood events in Europe. The following regions were analysed: a) German-Danish west coast b) Western facing coast of Sweden. c) Western facing coast in the Bothnian Sea d) West coast of the Baltic states e) West coast of Great Britain f) West coast of Ireland. Coordinates of those regions are given in Table 2.

We additionally inserted cross references to Fig. 6 where appropriate.

*In my option the discussion section is very good. However, I suggest considering whether readability would be improved if the discussion and conclusion were disentangled. The conclusion would only need to be a few short sentences addressing the objectives set out at the end of the introduction.*

We prefer to keep discussion and conclusion in one piece. We appreciate the suggestion and will consider it in the future.

*\*       L9 and elsewhere: Consider changing the text to make clear that "common driver" refers to a large scale metrological driver rather than a direct flooding driver like storm surge.*

Changes:
Line 9: Finally, we analyse if the observed compound extreme events had a common large-scale meteorological driver.

Line 17:  Drivers for floods are storm surges, waves, tides, precipitation, and high river discharge (Paprotny et al., 2020).

Line 73: Furthermore, we wanted to investigate if spatial patterns occur and if they are caused by one common meteorological driver.

Line 130: Rivers with this behaviour might indicate a common large-scale driver that causes extreme discharge and sea level at the same time.

Line 262: A common meteorological driver for compound flood events

Line 263: To see if the regions with a higher than expected number of compound flood events have a common large-scale meteorological driver we analysed the meteorological situation during these events.

Line 266: Our goal was to scrutinise whether large-scale compound flood events in these rivers have a specific Großwetterlage as their common meteorological driver.

Line 320: We identified the Großwetterlage Cyclonic Westerly as the common meteorological driver for the occurrence of large-scale compound flood events in North and Baltic Sea regions.

*       L10: The "than expected" phrase is a little ambiguous. I believe you mean more potential compound events are expected compared to simple random chance but it could be interpreted as meaning more events expected based on similar previous studies.

Changed Line 10:

The results of our investigation show that rivers along the western facing coasts of Europe experienced a higher amount of compound flood events than expected by pure chance.

*       L20: The sentence starting on this line requires a reference.

We added the citation to line 20:

The occurrence of extreme flood events either simultaneously or in close succession can lead to severe damage, which greatly exceeds the damage those events would cause separately (de Ruiter et al., 2020, Xue et al., 2022).

*       L21: "Zscheischler et al. (2018) described in further detail why it is essential to consider compound events for risk assessment." This is not very insightful.

Removed line 21 as suggested.

*       L27: "Potential compound flood events occur when large run-off from, e.g., heavy precipitation, leading to extreme river discharge, is combined with high sea level (storm surge)." So what type of compound event is this in terms of the Seneviratne et al., (2012) definition?

As stated above we removed the term "potential" to avoid confusion. Therefore, the compound event described by us is type 1 in terms of the definition by Seneviratne et al. (2012).

*       *L41: A lot of the listed studies focus on Asia/Oceana whereas much of the introduction concerns the U.S. and Europe. Kim et al. (2022) is a recent U.S. study that could be described here.*

We added the reference as suggested:

Studies have been conducted worldwide, with examples being the Zengwen River basin in Taiwan by Chen and Liu (2014), Shoalhaven River in Australia by Kumbier et al. (2018), Fuzhou in China by Lian et al. (2013), Dickinson Bayou watershed in Texas by Kim et al. (2022), and other locations in various countries.

*       *L43: I agree there is no established standards for detecting extreme events, however there are studies that compare the results obtained using different methods to identify extremes (e.g., Zheng et al. 2014) and changes in model set-up (e.g., Jane et al. 2022). There are also comparisons between findings from different studies (Ward et al. 2018; Ghanbari et al. 2021).*

We added to the end of Line 48:

Nonetheless, there have been some studies that investigated the sensitivity of their results regarding different methods of identifying extreme events (Zheng et al., 2014) and changes to the model set-up (Jane et al., 2022). Additionally, there are studies like Ghanbari et al. (2021) that compare their results to those of other studies, e.g., to Ward et al. (2018).

*       *L44: The items listed here are not equivalent. The percentile approach (inverse of return period) and events per year criteria are really two techniques amongst many for choosing a threshold in the peaks over threshold approach. Univariate extremes are typically identified using either a peak over threshold or block maxima approach. In terms of a bivariate sample, univariate extremes of a variable can be paired with concurrent or near concurrent values of a second variable in a procedure referred to as one-way conditional sampling (Moftakhari et al. 2019). In two-way conditional sampling, the procedure is repeated conditioning on the other variable producing two conditioned samples (Ward et al. 2018). Zheng et al. (2014) discusses the advantages and limitations of three methods of identifying bivariate extremes.*

We thank the reviewer for their detailed information on bivariate sampling. We believe that this additional information would rather confuse the reader if added to the publication, since it is not used.

However, we modified Line 45:

For example, the thresholds for extreme events were calculated by utilising the return period (Bevacqua et al., 2019), a certain number of events per year (Hendry et al., 2019; Ganguli et al., 2020), or utilizing a percentile approach (Paprotny et al., 2018a). Other studies chose block maxima to detect extreme events (Engeland et al., 2004).

*       L51: Bevacqua et al. (2020) has some predictions regarding the high emissions scenario for Europe. Also, you need to state that flood damage amounting to a considerable proportion of some countries GPD by the end of the century is a prediction at this stage!*

We added the reference as suggested to Line 55:

Bevacqua et al. (2019, 2020) reported a strong increase in the occurrence rate of compound flooding events for the future, especially for northern Europe, mainly due to the stronger precipitation as the result of a warmer atmosphere carrying more moisture.

In addition, we modified line 51:

Feyen et al. (2020) projected that in case of a high emissions scenario, the damages caused by floods will represent a considerable proportion of some country's national gross domestic product (GDP) at the end of the century.

*       L53: Perhaps discuss the results of the Norway study and remove the sentence about different spatial scales as this feels a bit like repetition. Bermúdez et al. (2021) is a regional study which could be cited here.*

We believe that it is not target-oriented to discuss the results of the Norway study by Poschlod because it focuses on rain on snow and rain on saturated soil.

We added the reference to line 53 as requested:

Studies that investigated the effect of climate change on compound flood events focused on various regions of interest, for example, Bevacqua et al. (2019) on entire Europe, Poschlod et al. (2020) on Norway, Bermúdez et al. (2021) on the rivers Mandeo and Mendo in Spain, and Ganguli et al. (2020) on northwestern Europe.

*       L63: The text states that copulas "rely on a huge amount of data points", given that copulas are widely applied in climate studies the following sentence appears to be a contradiction where it states that "the amount of available data points is always insufficient for this kind of analysis with many studies operating at merely 30 extreme events". Changing the first sentence to state that a large number of data points may be required for the copula fit to be robust will remove the apparent contradiction. Jane et al. (2022) explores the sensitivity of copula family to sample size.*

The requested changes were made and can be seen in the general comments section above.

*       L85: A few automatic threshold techniques could be cited here.*

We added citations as requested:
There are ways to use automatic threshold approaches for detecting extreme events, like goodness of fit p-value (Solari et al., 2017) or the characteristics of extrapolated significant wave heights (Liang et al., 2019), but they struggle due to the diverse characteristics in the time series of drivers that cause coastal floods (Camus et al., 2021).

*       L86: I would be really interested to read studies where local properties, like flood protection or elevation of the surrounding area play a role in threshold selection.*

We agree but we are also not aware of such studies.

*       *L94: Compound flooding studies such as Ward et al. (2018) where a percentile threshold is applied over a large spatial scale should be noted here.*

We added the reference as requested:
We, therefore, chose the peaks-over-threshold (Pickands III, 1975) method to select extreme events by using percentiles, like in the works of Rantanen et al. (2021), Fang et al. (2021), Lai et al. (2021), Ward et al. (2018), and Ridder et al. (2018).

*       *L97: I suggest requirement for independent events be discussed before describing the variation in the number of (independent) extreme events with river characteristics.*

As suggested, we moved the paragraph starting in line 104 to line 95.

*       *L106: The definition of separated is not clear here. Is it separated by four consecutive days containing no threshold exceedances?*

This is correct. We improved the wording to:
This means that two events are considered to be separate if the threshold is not exceeded for four consecutive days, like in Haigh et al. (2016).

*       *L108-109: I do not understand the point being made in this very short paragraph. "In order to enable a good comparison between different rivers, the number of extreme discharge and sea level data points should be the same for all of them." I don't think the number of extreme discharge and sea level points necessarily need to be the same for a comparison to be valid. Furthermore, with this rational is there not a danger of choosing a threshold that permits events that are not extreme so a comparison can be made?*

To make our point clearer we modified the text as follows:

In order to enable a good comparison between different rivers, the number of extreme discharge and sea level data points should be the same for all of them. Then again, extreme events should be rare by definition, regardless of the river size, therefore only occurring scarcely throughout the year. This especially prevents the accidental analysis of events that are normally not considered as extreme. To test the influence of the extreme event definition on possible patterns, […].

*       *L114: Please define more clearly which events have an "average return period of 0.5 years for extreme events".*

We changed line 114:

Moreover, the threshold tuning results in an average return period of 0.5 years for extreme discharge and sea level events since the return period can be defined as […].

*       *L130: Previous studies which implement a similar procedure should be cited e.g., Svensson and Jones (2002), Zhang et al. (2013) and Nasr et al. (2021). Couasnon et al. (2020) presents a similar approach to the one proposed in this paper that uses the binomial distribution to assess the number of annual maxima storm surge – river discharge co-occurrences expected under the independence assumption.*

As suggested, we added the following to line 130:

Other studies in the past also utilised data permutation, see, for example, Svensson and Jones (2002), Zheng et al. (2013), and Nasr et al. (2021).

*       L136: Couasnon et al. (2020) speaks to this.

We added the following to line 138:

This was similarly stated by Couasnon et al. (2020).

*       L132: "east coast of Great Britain exhibits a low number of compound flood events" This is repetition of the previous sentence.

We removed the repetitive part from line 231 as suggested:
The east and south facing coasts of the Bothnian Bay, Bothnian Sea and the Skagerrak show the lowest frequency of compound flood events. Similarly, the east coast of Great Britain exhibits a low number of compound flood events, in contrast to the west coast.

*       L161: Water levels at the mouth of a river are often heavily influenced by sea level (e.g., Moftakhari et al. 2019), and therefore are not desirable in this context since they are not a reliable measure of the river discharge caused by inland rainfall.

Our results are based on simulated discharges at the river mouth, not on observed discharges. The HD model does not consider sea level impact; therefore, it does not cause problems in our analysis.

*       L228: What time period?

We added information on the time period to line 228:

A total of 26% of the rivers along the coasts had eight or more compound flood events during the time period 1995-2018.

*       Figure 1: The choice of color bar makes this figure difficult to interpret. Consider using a cold to hot color scale to more easily identify hotspots (and cold spots).

Changed colormap of Figure 1 to viridis:

[Figure]

*       L241: Could add "and are discussed in the following subsections" to the end of this paragraph to guide the reader.

Added as suggested to line 241:

Some images for these tests are in Appendix A for the sake of readability, and they are discussed in the following subsections.

*       L248: What pattern are you referring to here?

We modified line 248:

The pattern of western facing coasts having a higher number of compound flood events than expected by random sampling remained persistent throughout different time periods, even though it is somewhat more pronounced in the more recent one.

*       L260: Here and elsewhere it would be interesting to explore possible causes of these differences.

We added the following information to line 329:

There was a certain amount of variation in the pattern, which can be attributed to randomness and the different setups. Due to the limited number of compound flood events, even small variations to their definition, like changes in the allowed lag, have a minor influence on the results.

*       L284: So what weather systems were responsible for the rest of the compound events in Great Britain?

We added the following information to line 284:

[revised manuscript text omitted]

---

## Author Response (AR2)

**Reply to reviewer's comments**

We thank the reviewer for the thorough reading of the manuscript and the valuable remarks that helped us to improve the manuscript. In the following, the original reviewer comments are given in italic and all line numbers and figure numbers refer to the submitted version.

**Reply to review of reviewer 2**

*As reviewer 2 in the last round of the review process, I would like to congratulate the authors on the revised manuscript. In my opinion adding a separation criterion to the de-clustering procedure has increased the robustness of the method, while paragraph structure and quality of the Figures are much improved on the initial submission.*

We thank reviewer 2 for the overall positive evaluation of our revised manuscript and very detailed comments.

*In my view, there are several issues that need to be addressed before the manuscript is ready for publication. Overall, the methods and analysis are robust, however the purpose of the work/ utility of the results remains unclear, with only a single possible useful application alluded to at the very end of the manuscript. The explanation of the statistics in the introduction requires attention and the discussion of existing studies could be revised to provide more insight.*

We address these concerns in the following sections.

*The authors really need to figure out the overall aim/purpose of the work and state it before the three tasks are introduced in the abstract. A reader needs to know why it is worth their time reading the paper. In the current text the aim of the work is too vague. For example, "It is important to" is subjective, and what constitutes a underlying mechanisms is not clear. "Our study focuses on the analysis of potential compound flood events" I could guess this from the previous sentences! "with the following contributions" they are steps rather than contributions.*

We made the following changes to the abstract starting from line 2:

Compared to the occurrence of single extreme events, co-occurring or compound extremes may substantially increase risks. To adequately address such risks, improving our understanding of compound flood events in Europe is necessary and requires reliable estimates of their probability of occurrence together with potential future changes. In this study compound flood events in northern and central Europe were studied using a Monte-Carlo based approach that avoids the use copulas. […]

Furthermore, we made the following addition to line 407:

This leads to the question of how the frequency of compound flood events might change for the various parts of Europe, which is vital for regional coastal adaptation.

*The first paragraph in the introduction zeros in on coastal flooding, however later in the introduction studies concerning "compound inland floods" are discussed. Due to the depth of the literature on compound flooding, I suggest omitting studies that do not involve discharge-*

*surge compounding.*

We removed the three references in line 36 in favour of a study that involves discharge-surge:

Several studies conducted over the last years have shown the importance and catastrophic nature of compound flood events for various locations.

One example is the flooding of Jacksonville (Florida) where the surge caused by the strong winds of Hurricane Irma stalled the fluvial discharge (Juarez et al., 2022).

*Phases such as "many studies" are used rather a lot in the introduction. The introduction should be crafted to describe the relevant findings of the previous studies in a way that engages the reader. Lines 60-63 are a good example of where the key findings of the individual studies should be stated rather than summarized with a semi-relevant generic statement two sentences later. Furthermore, it is not true that results cannot be compared, its just that differences may at least partially be down to the chosen approaches. The paragraph starting on L64 is a well-crafted thoughtful paragraph.*

We reduced the number of listed citations and further elaborated on the remaining ones.

Line 39:

Moreover, several studies have been conducted on a larger spatial scale in Europe. Considering data from 1901-2014 and gauges from northwestern Europe Ganguli and Merz (2019) found opposing trends in the magnitude of compound flood events depending on the latitude of the gauge. They reported increases at midlatitudes (47°N to 60°N) and decreases for gauges at high latitude (>60°N). Svensson and Jones (2002) analysed the dependence of high sea surge, river flow, and precipitation in the UK. They found a higher number of compound flood events on the western than on the eastern cost, while Paprotny et al. (2020) demonstrated that hydrodynamic models are capable of identifying real world compound flood events in north-western Europe.

Line 49:

Additionally, there have been studies modelling compound flood events in rivers on a local scale such as for the Zengwen River basin in Taiwan by Chen and Liu (2014), the Shoalhaven River in Australia by Kumbier et al. (2018), and the Fuzhou in China by Lian et al. (2013).

Line 60:

Nonetheless, there have been some studies that investigated the sensitivity of their results. Zheng et al. (2014) compared three classes of statistical methods and found that the point process method overestimated the dependence of extremes while the conditional method underestimated it. In a similar vein, Jane et al. (2022) assessed that their estimates of the potential for compound events were highly sensitive to the statistical model setup.

Removed sentence at the end of line 61.

Additionally, line 53 was modified:

A direct comparison of the results from different studies is hampered by the use of different approaches, data, analysis periods, and other factors.

*Most of the statements relating to the statistical modeling can be made clearer and some are erroneous. "They were introduced in Sklar (1959)" is an example of the latter. I suggest consulting with a statistician familiar with this field to tighten up the text regarding the copula modeling. A definition of a copula is also missing.*

We tried to present the arguments with more rigour and references to existing studies. The revised paragraph reads as follows:

Many studies utilised multivariate extreme value theory and copulas to describe the data distribution of two or more time series and investigate the dependence between extreme events (Hao et al., 2018). In climate research, the amount of available data points is often very small, with many studies operating at merely 30 extreme events. This can cause large uncertainties when trying to evaluate the tail dependence of the multivariate distributions (Serinaldi, 2013; Serinaldi et al., 2015; Joe, 2014). An alternative approach is based on Monte-Carlo based simulations where the dependence between joint extremes is studied by randomly rearranging one of the time series. Given our small sample size, in the following we used such an approach to avoid the uncertainties associated with the use of copulas in small samples.

*\*        L18: "Drivers for floods are storm surges, waves, tides, precipitation, and high river discharge" Perhaps worth noting that stretched of river where these drivers combine to exacerbate flooding are referred to as transition zones (e.g., Bilskie and Hagen 2018).*

We added this information to line 19:

The area of the river in which two or more of these drivers influence the water level are called flood transition zones (Bilskie and Hagen, 2018).

*\*        L29: "Compound flood events occur when large run-off from, e.g., heavy precipitation, leading to extreme river discharge, is combined with high sea level (storm surge)" there are many types of compound flood event, however I agree this is a good place to state that this is the type of compound flood the work will focus on.*

We added this information to line 29:

This study focuses on compound flood events that occur when large runoff from, e.g., heavy precipitation, leading to extreme river discharge, is combined with high sea level (storm surge).

*\*        L31: "In the following text we will note "potential compound flood events " as "compound flood events" for the sake of readability, with regards to literature see, e. g., Ganguli and Merz (2019), Jane et al. (2020), or Couasnon et al. (2020)." I suggest retaining this text!*

Added the sentence as suggested with an additional explanation as suggested in the first review.

Local flood protection and topography might prevent compounding extreme events from causing floods. Due to the size of the study area, we cannot take this into account and will

denote these "potential compound flood events" as "compound flood events" in the following text for the sake of readability.

\* L33: "The occurrence of extreme flood events either simultaneously or in close succession can lead to severe damage, which greatly exceeds the damage those events would cause separately (de Ruiter et al., 2020; *Xu et al., 2022)." Most of the subsequent studies you describe only have one flood event caused by multiple drivers arising simultaneously.*

We changed the sentence in L33 to:

The occurrence of extreme flood and surge events either simultaneously or in close succession can lead to severe damage, which greatly exceeds the damage those events would cause separately (de Ruiter et al., 2020; Xu et al., 2022).

*\* L43: "All of them found that the assumption of independence between drivers leads to a huge underestimation of the occurrence rate of compound events." This is not true for all locations.*

Changed line 43 to:

Many studies found that the assumption of independence between drivers leads to an underestimation of the occurrence rate of compound events.

*\* L56-57: I believe these are all peaks-over-threshold approaches. There are also different peaks-over-threshold de-clustering methods e.g. storm window or runs method.*

We added the following information to line 118 since we feel that this is a better location for this information:

A critical element in the analysis is the definition of a de-clustering window such that subsequent events can be considered as independent. A frequently used window size is based for example on the typical duration of storms in the area (e.g. Harley, 2017; Camus et al., 2021). Here, we chose a de-clustering time of three days as used in other studies spanning larger domains (e.g., Bevacqua et al. (2019); Ward et al. (2018); Haigh et al. (2016)).

*\* L82: "Serinaldi et al. (2015) therefore concluded that those results are "highly questionable and should be carefully reconsidered". This is a very general statement, does he really say all copula models are highly questionable and should be carefully reconsidered.*

*The paragraph starting on L84 ends with "Consequently, we chose to study compound flood events by using a methodology that does not utilise copulas". It is strange because copulas are not mentioned at all in the rest of the paragraph. Copulas are not needed to calculate tail dependence.*

Please see our changes made in the "general comments" section since we rewrote this paragraph.

*\* L129: River discharge is traditionally denoted by a Q.*

We changed line 129 accordingly:

For the discharge of rivers we chose the 90th percentile $Q_{90}$ and for the sea level the 99th percentile $S_{99}$.

*       L134: As I stated in my last review, I do not believe this is true. The trade-off between a large sample whilst ensuring the sample only contains actual extremes could be incorporated into the previous paragraph and then L134-137 could be deleted.

We removed the sentence starting on line 134. We moved the sentences in lines 135-137 to the beginning of the paragraph.

Extreme events should be rare by definition, regardless of the river size, therefore only occurring scarcely throughout the year. This especially prevents the accidental analysis of events that are normally not considered as extreme. On the other hand, the choice of our threshold needed to take the limited data availability into account. Hence, we were forced to choose our thresholds low enough to ensure that enough points were available for robust statistical analysis.

*       L168: "This was similarly stated by Couasnon et al. (2020)." You can just cite Couasnon et al. (2020) at the end of the previous sentence.

Moved the reference to the end of the previous sentence in line 168:

As a result, we would see a much lower number of compound flood events in the non-randomised data; therefore suggesting a false dependence (Couasnon et al., 2020).

*       L190-195: Please refer to my comment in the last round of reviews regarding the optimum location of river gauges for this type of study.

Based on our reply to your comment we added the following information to line 197:

For our analysis, we utilised several model-based data sets which varied in forcing, regions and time frames.

The following sentence was added to line 198:

The simulated discharges are solely caused by the atmospheric forcing and the hydrological processes over land. The influence of the sea level on discharge in the estuaries of the rivers is not considered so that this influence (e.g., *Moftakhari et al. 2019*) does not cause problems in the determination of river floods.

*       L294 & 219: In English the abbreviation "vs." is typically shorthand for "verses". "v." is generally used for "version".

Changed line 219:

Using E-OBS v. 22, HydroPy was driven by daily temperature and precipitation at 0.1° resolution from 1950–2019.

We assume that the reviewer meant line 204 instead of 294:

The HD model v. 5.0 (Hagemann and Ho-Hagemann, 2021) was set up over the European domain covering the land areas between -11° W to 69° E and 27° N to 72° N at a spatial resolution of 5 min (ca. 8-9 km).

*	*L223: Grammar. This could work: "… found precipitation data from ERA5 to be of a higher quality than from EOBS."*

Changed line 223 as suggested:

Investigations by Rivoire et al. (2021) found precipitation data from ERA5 to be of higher quality than from E-OBS.

*	*L266: Quite wordy. I would remove the reference to the eastern British coast in this sentence as it is discussed in the following sentence.*

Removed the reference in line 266 as suggested:

The east and south facing coasts of the Bothnian Bay and Bothnian Sea in the Baltic Sea, as well as Skagerrak, show the lowest frequencies of compound flood events.

*	*L270: Doesn't this plot simply reflect the number of compound events in the record i.e., Figure 2. The west coast of France has a small number of compound events at most sites, therefore we'd expect it to be within 2 standard deviations. What about the other coasts?*

We agree that a very high number usually reflects being outside of 2 standard deviations. Nonetheless, the number of events depends to a certain extent also on the amount of data points. Long discharge events therefore offer a higher chance by randomness to coincide with a sea level extreme event. Analysing if the number of events is within 2 standard deviations is therefore a safeguard.

*	*L271: Remove "or outside" it is superfluous.*

Removed "or outside" as suggested from line 271:

Utilising our randomisation method (cf. Sect. 2) yielded Fig. 3 that shows if the amount of observed compound flood events for each river is within the $2\sigma$ interval produced by the randomised data sets.

*	*Figure 2 (caption): "The number of discharge and sea level extreme events was limited to two events per year." That is on average.*

Added to caption of Figure 2:

The number of discharge and sea level extreme events was limited to two events per year on average.

*	*L280: The term "pattern" is ambiguous here.*

Changed line 280 to:

The overall pattern indicating that western coasts have the tendency of showing more events than expected by pure chance remains stable throughout these different data set combinations.

*       L283: Change "sections" to "periods".

Changed "sections" to "periods" as suggested in line 283:

Next, we split the ECOSMO–coastDat3 and HD5–EOBS data into two 30 year periods, from 1960 to 1989 (Fig. 4b) and from 1990 to 2019 (Fig. 4c).

*       L321: Grammar. "flood compound events".

Changed the order of words in line 321:

Anticyclonic Westerly is known to lead to precipitation in the area of the Baltic countries (Jaagus et al., 2010), which in combination with the south-eastern wind direction are responsible for around a third of the compound flood events in the Baltic and western facing Finnish area, due to the orientation of their coastline.

*       L342: Remove "such".

Removed "such" from line 342:

Furthermore, the rivers were coloured red if the number of compound flood events is above the 2σ interval of randomised sea level data, blue if below the interval, and grey otherwise, as in Fig. 3.

*       L376: "In addition, we demonstrated that there exists a correlation between river catchment size and the number of compound flood events. It can be seen that, regardless of the estuary orientation, the number of compound flood events declined with increasing catchment size" All the information in the first sentence is contained within the next sentence. Also, they declined "on average", there is not an exact relationship.

We combined both sentences and added the information that it declined "on average":

In addition, we demonstrated that regardless of the estuary orientation, the number of compound flood events declined on average with increasing catchment size.

*       L378: "The reason for this might be that rivers with smaller catchment areas are capable of reacting faster to precipitation that appears during the storm events, which also causes the storm surges." Did accounting for the catchment response times by including a lag weaken the trend?

We thank the reviewer for the interesting suggestion. As stated in the publication text, we used a constant lag. We are aware that studies like Ganguli and Merz 2019 used a lag time that depended on the catchment size (formula 1 in their paper). Our concern is that especially

for large rivers the lag would massively depend on the location of the precipitation. For example, the discharge of the Elbe would react much faster to precipitation in Hamburg than to precipitation near Prague. Therefore, it would be very challenging to quantify the lag for all the rivers in our study with a detailed investigation being beyond the scope of this work. Nevertheless, it is a very compelling topic that could be considered in future studies.

We therefore added to Line 395:

They could also attempt to quantify the lag for each catchment individually, which is currently challenging for large rivers since their lag depends on the location of the precipitation.

Reference:
Ganguli, P. and Merz, B.: Trends in compound flooding in northwestern Europe during 1901–2014, Geophysical Research Letters, 46, 10 810–10 820, https://doi.org/10.1029/2019GL084220, 2019.

*	L388: Maybe change "for design events to test flood protection structures" to "such as design events used to assess the level of protection afforded by flood defence structures."

Changed line 388 as suggested by the reviewer:

The lack of a parametric model impedes the possibility of deriving engineering quantities such as design events used to assess the level of protection afforded by flood defence structures.

*	L389: "ensemble data" From climate models I assume. Another benefit to ensemble data is the higher spatial resolution. A drawback that it is numerically derived rather than observed.

Changed line 389 to:

Future work can further examine these findings by using ensemble from climate models data over a longer time frame, e.g. 50 years and more.

Additionally, we added to line 393:

One potential drawback is the reliance on the capabilities of numerical models to adequately generate those compound extreme events.
* * *
List of added references:

Bilskie, M. V., & Hagen, S. C. (2018). Defining flood zone transitions in low-gradient coastal regions. Geophysical Research Letters, 45, 2761– 2770. https://doi.org/10.1002/2018GL077524

Hao, Z., Singh, V. P., and Hao, F.: Compound extremes in hydroclimatology: a review, Water, 10, 718, https://doi.org/10.3390/w10060718, 2018.

Juarez, B., Stockton, S. A., Serafin, K. A., and Valle-Levinson, A.: Compound flooding in a subtropical estuary caused by Hurricane Irma 2017, Geophysical Research Letters, 49, e2022GL099 360, https://doi.org/10.1029/2022GL099360, 2022.

Moftakhari, H., Schubert, J. E., AghaKouchak, A., Matthew, R. A., and Sanders, B. F.: Linking statistical and hydrodynamic modeling for compound flood hazard assessment in tidal channels and estuaries, Advances in Water Resources, 128, 28–38,535 https://doi.org/10.1016/j.advwatres.2019.04.009, 2019.

Serinaldi, F.: An uncertain journey around the tails of multivariate hydrological distributions, Water Resources Research, 49, 6527–6547, https://doi.org/10.1002/wrcr.20531, 2013

---

## Author Response (AR3)

**Reply to reviewer's comments**

We thank the reviewer for their thorough reading of the manuscript and their valuable remarks that helped us to improve the manuscript. In the following, the original reviewer comments are given in italic and all line numbers and figure numbers refer to the original submitted version that was reviewed if not mentioned otherwise.

**Reply to review of reviewer 2**

*Thank you for addressing all my comments. The paper is a very worthy addition to the compound flooding literature- congratulations!!*

We thank reviewer 2 for all the suggestions made which helped us to improve the manuscript.

\* *L19: Remove "flood".*

We removed "flood" as suggested.

The area of the river in which two or more of these drivers influence the water level are called transition zones (Bilskie and Hagen, 2018).

\* *L31: To me the "flood" is the response variable. I proffer "discharge and storm surge events" is more accurate.*

We changed line 31 accordingly:

The occurrence of extreme discharge and storm surge events either simultaneously or in close succession can lead to severe damage, which greatly exceeds the damage those events would cause separately (de Ruiter et al., 2020; Xu et al., 2022).

\* *L71: I feel there is a sentence missing here briefly explaining how the randomization test worsk after re-arranging the time series.*

We assume that the reviewer intended to write "L81" and made the following changes starting from line 80:

For this, we randomised our datasets in a bootstrap process and investigated the number of compound extreme events in them, which resulted in a probability distribution in case of independence. Rivers with a number of observed compound extreme events outside of the 95% confidence interval of two standard deviations might have a common large-scale driver.

\* *L73: Remove "based".*

Removed "based" as suggested from line 73.

An alternative approach is based on Monte–Carlo simulations where the dependence between joint extremes is studied by randomly rearranging one of the time series.

*        L115: Is this more specific that it is a "lower number of independent extreme events for a specified quantile threshold". And so the next sentence is "Smaller rivers, however, usually have rather short extreme events, and consequently a larger number of independent extremes for the same quantile threshold."*

We incorporated the suggested changes to the sentences starting in line 114:

Large rivers like the Elbe show the tendency of having very long extreme events that can last for several weeks, therefore resulting in a lower number of independent extreme events for a specified quantile threshold. Smaller rivers, however, have usually rather short extreme events, and consequently a larger number of independent extremes for the same quantile threshold.

*        L267: Grammar. "... remained persistent throughout ...". I suggest removing "remained" as it is superfluous here.*

We changed "remained" in Line 267:

The pattern of western facing coasts having a higher number of compound flood events than expected by random sampling is persistent throughout different time periods, even though it is somewhat more pronounced in the more recent one.

*        L268: "This is seen by the generally higher number of rivers above the 2σ interval, indicating that compound flood events can potentially occur in these months." This sentence does not make sense. What months are you talking about. Be sure to check that you're not just repeating the previous sentence.*

We decided to remove this sentence since it is repeating the information of the previous one.

*        L274: Remove")".*

Removed ")" from the previous sentence in line 274:

As a first test, we changed the lag from zero to three days which is shown in Fig. 4d.

*        L305: Could this also be because different Großwetterlage lead to similar climatic conditions in Ireland because it is so far from Germany i.e. the location where the weather types are derived for.*

According to the KATALOG DER GROSSWETTERLAGEN EUROPAS (1881-2009) [roughly translates to "Catalogue of the Großwetterlagen of Europe"] by Werner and Gerstengarbe (2010), the Großwetterlagen are defined over a large domain that includes Ireland. An example of the domain can be seen on page 120 of the following document. Furthermore, the Großwetterlagen are not specifically derived for Germany.

Link to Document: https://www.pik-potsdam.de/en/output/publications/pikreports/.files/pr119.pdf

*       *L364: Grammar: Consider changing to "using ensembles from climate models that cover longer time frames, e.g. 50 years or more."*

Future work can further examine these findings by using ensembles from climate models that cover longer time frames, e.g. 50 years or more.